# A Deep Insight into the Ion Foreshock with the help of Test-particle Two-dimensional Simulations

Philippe Savoini[1] and Bertrand Lembège[2]

[1]LPP (Laboratoire de Physique des Plasmas), Ecole Polytechnique, Route de Saclay, 91128, Palaiseau, France
[2]LATMOS (Laboratoire Atmosphères, Milieux, Observations Spatiales, IPSL/CNRS/UVSQ, 11 Bd d'Alembert, 78280, Guyancourt, France

**Correspondence:** P. Savoini (philippe.savoini@sorbonne-universite.fr)

**Abstract.** Two dimensional test-particle simulations based on shock profiles issued from 2D full PIC simulations are used in order to analyze the formation processes of ions backstreaming within the upstream region after these interact with a quasi-perpendicular curved shock front. Two different types of simulations have been performed based on (i) a "*FCE*" (Fully Consistent Expansion) model which includes all self-consistent shock profiles at different times, and (ii) a "*HE*" (Homothetic Expansion) model where shock profiles are fixed at certain times and artificially expanded in space. The comparison of both configurations allows to analyze the impact of the front non stationarity on the backstreaming population. Moreover, the role of the space charge electric field $E_l$ is analyzed by ~~switching it in/off~~ including or cancelling the $E_l$ component in the simulations. A detailed comparison of these two last different configurations allows to show that this $E_l$ component plays a key role in the ion reflection process within the whole quasi-perpendicular propagation range. Simulations evidence that the different "*FAB*" (Field-Aligned Beam) and "*GBP*" (Gyro-Phase Bunch) populations observed *in-situ* are essentially formed by a $\overrightarrow{E}_t \times \overrightarrow{B}$ drift in the velocity space involving the convective electric field $\overrightarrow{E}_t$. Simultaneously, the study emphasizes (i) the essential action of the magnetic field component on the "*GBP*" population (i.e. mirror reflection~~and Fast Fermi acceleration~~) and (ii) the leading role of the convective field $\overrightarrow{E}_t$ on the "*FAB*" energy gain. ~~the leading role of the electrostatic (longitudinal) field $\widetilde{E}_t$ built up within the shock front in the acceleration process in addition to the magnetic mirror reflection (Fast Fermi).~~ In addition, the electrostatic field component $\overrightarrow{E}_l$ appears as essential for ~~form~~ reflecting ions at high $\theta_{Bn}$ angles and in particular at the edge of the ion foreshock around 70°. Moreover, the "*HE*" model shows that the rate $BI_\%$ of backstreaming ions is strongly dependent on the shock front profile which varies because of the shock front non stationarity. In particular, reflected ions appear to escape periodically from the shock front as "bursts" with an occurrence time period associated to the self-reformation of the shock front.

## 1 Introduction

While upstream ions of the incoming Solar Wind interact with the curved terrestrial bow shock, a certain percentage is reinjected back into the solar wind and propagates along the interplanetary magnetic field (*IMF*): they form the so-called ion foreshock. This population has been extensively studied both with the help of experimental data (Tsurutani and Rodriguez, 1981;
Paschmann et al., 1981; Bonifazi and Moreno, 1981a, b; Fuselier, 1995; Eastwood et al., 2005; Oka et al., 2005; Kucharek, 2008; Hartinger et al., 2013) and numerical simulations (Blanco-Cano et al., 2009; Lembege et al., 2004; Savoini et al., 2013; Kempf et al., 2015; Savoini and Lembège, 2015; Otsuka et al., 2018).

Even if we restrict ourselves to the quasi-perpendicular region (i.e. for $45^o \leq \theta_{Bn} \leq 90^o$, where $\theta_{Bn}$ is the angle between the local shock normal and the *IMF*), different types of backstreaming ions are identified ~~are observed~~: (a) some are characterized by a gyrotropic velocity distribution and form the field-aligned ion beam population (hereafter "*FAB*"), and conversely (b) others exhibit a non-gyrotropic velocity distribution and form the gyro-phase bunched ion population (hereafter "*GPB*"). None of these populations has yet a well established origin and different mechanisms have been proposed for years (Möbius et al., 2001; Kucharek et al., 2004): *(i)* scenarii based on the specular reflection (Sonnerup, 1969; Paschmann et al., 1980; Schwartz et al., 1983; Schwartz and Burgess, 1984; Gosling et al., 1982) with or without the conservation of the magnetic moment, *(ii)* scenarii which invoke the leakage of some magnetosheath ions producing low energy *FAB* population (Edmiston et al., 1982; Tanaka et al., 1983; Thomsen et al., 1983). Nevertheless, the origin of "*FAB*" ions could be due to *(iii)* the diffusion of some reflected ions (called "*gyrating ions*"; these ions are reflected by the supercritical shock front but do not manage to escape into the upstream region and go into the downstream region after their initial gyration (Schwartz et al., 1983)). The diffusion can be generated by upstream magnetic fluctuations (Giacalone et al., 1994) or more directly by the shock ramp itself (with a pitch angle scattering during the reflection process) (Kucharek et al., 2004; Bale et al., 2005). All scenarii have some drawbacks and are not able to explain clearly the origin of both populations. On the other hand, "*GPB*" are preferentially observed at some distances from the curved shock front (Thomsen et al., 1985; Fuselier et al., 1986a) and their synchronized nongyrotropic distribution comes as a part of a low-frequency monochromatic waves trapping (Mazelle et al., 2003; Hamza et al., 2006), or of beam-plasma instabilities (Hoshino and Terasawa, 1985). As a conclusion, it is quite difficult to discriminate between these different scenarii which can be present simultaneously or separately in time.

Our previous papers (Savoini et al., 2013; Savoini and Lembège, 2015) were focused on the origin of these two populations. A large scale two-dimensional PIC simulation of a curved shock has been used, where full curvature and time-of-flight effects for both electrons and ions are self-consistently included. Our simulations have shown that both "FAB" and "GPB" populations and their typical associated pitch angle distributions observed experimentally (Fuselier et al., 1986b; Meziane, 2005) have been retrieved not far from the front (until to $2-3R_E$ where $R_E$ is the Earth's radius). Moreover, results have shown that these two populations can be generated directly by the macroscopic electric $\overrightarrow{E}$ and magnetic $\overrightarrow{B}$ fields present at the shock front itself. In other words, the differences observed between "*FAB*" and "*GPB*" populations are not the result of distinct reflection processes but are the consequence of the time history of ions interacting with the shock front: "*FAB*" population loses their initial phase coherency by suffering several bounces along the front, in contrast with the "*GPB*" population which suffers mainly one bounce

(i.e. mirror reflection process). This important result was not expected and greatly simplifies the question on each population origin (Savoini and Lembège, 2015)

Nevertheless, some further questions still need to be answered which are difficult to investigate with full PIC simulations (because of the self-consistency) in order to analyze several aspects of the reflection process. For this reason, we use herein complementary test-particle simulations to clarify the respective impact of the shock curvature and the time variation of the

macroscopic fields at the shock front on the backstreaming ion reflection process. The main questions presently addressed are summarized as follows:

  1. Is the reflection process non-continuous in time (burst-type reflection process) or not ? In this case, how is it linked to the $\theta_{Bn}$ angle variation (i.e., space dependence) and/or to the shock profile variation (i.e. time dependence) ?

  2. What is the impact of the space charge electric field localized within the shock ramp on the reflection process ?

3. What kind of reflection mechanisms can be identified in present simulations ?

The paper is organized as follows. Section 2 briefly summarizes the conditions of the previous 2D PIC simulations (Savoini and Lembège, 2015) and of present particle test simulations. In sections 3 and 4 results of test particles are presented and the ion reflection processes are investigated. Discussion and conclusions will be presented in section 5 and 6, respectively.

## 2 Numerical simulation conditions

The numerical conditions concerned in the present paper are similar to those described in Savoini et al. (2013) and Savoini and Lembège (2015). In short, we used a 2D dimensional, fully electromagnetic, relativistic particle code based on standard finite-size particle technique (similar to Lembege and Savoini (1992, 2002) for planar shocks).

### 2.1 Self-consistent full PIC Simulations

The code solves Maxwell and Poisson's equations in the Fourier space (so called pseudo-spectral code) which allows to

separate the electric field contribution in two distinct parts : (i) a "longitudinal" or electrostatic component hereafter denoted by a subscript "$l$" (built up by the space charge effects $\overrightarrow{\nabla} \overrightarrow{E_l} = \rho/\epsilon_o$) and (ii) a "transverse" or induced component hereafter denoted by a subscript "$t$" (coming from the temporal variations of the magnetic field $\overrightarrow{\nabla} \times \overrightarrow{E}_t = -\partial \overrightarrow{B}/\partial t$ ) ~~and then, fields are separated into transverse electromagnetic components (induced electric field), hereafter denoted by a subscript "t", and longitudinal electrostatic components hereafter denoted by a subscript "l" (space charge effects).~~ The longitudinal component is

essentially built up within the shock front due to the different dynamics of ions and electrons, whereas the induced component is mainly generated by the propagating shock front itself (see Figure 1 Panel 2a) through the convective term $\overrightarrow{E}_t = -\overrightarrow{U} \times \overrightarrow{B}$ ~~at the shock~~ (where $\overrightarrow{U}$ corresponds to the bulk shock front velocity since we are in the Solar Wind frame). In addition, the subscripts $\parallel$ and $\perp$ stand for parallel and perpendicular directions to the local magnetic field, respectively. In Figure 1 and followings, the $X - Y$ reference frame is the Solar Wind frame with the third direction along $Z$ pointing backward into the

plot. Then, $\overrightarrow{E}_t$ has the direction of the increasing $Y$ and $\overrightarrow{\nabla} B$ has the same direction as a the present $\overrightarrow{U}$ vector.

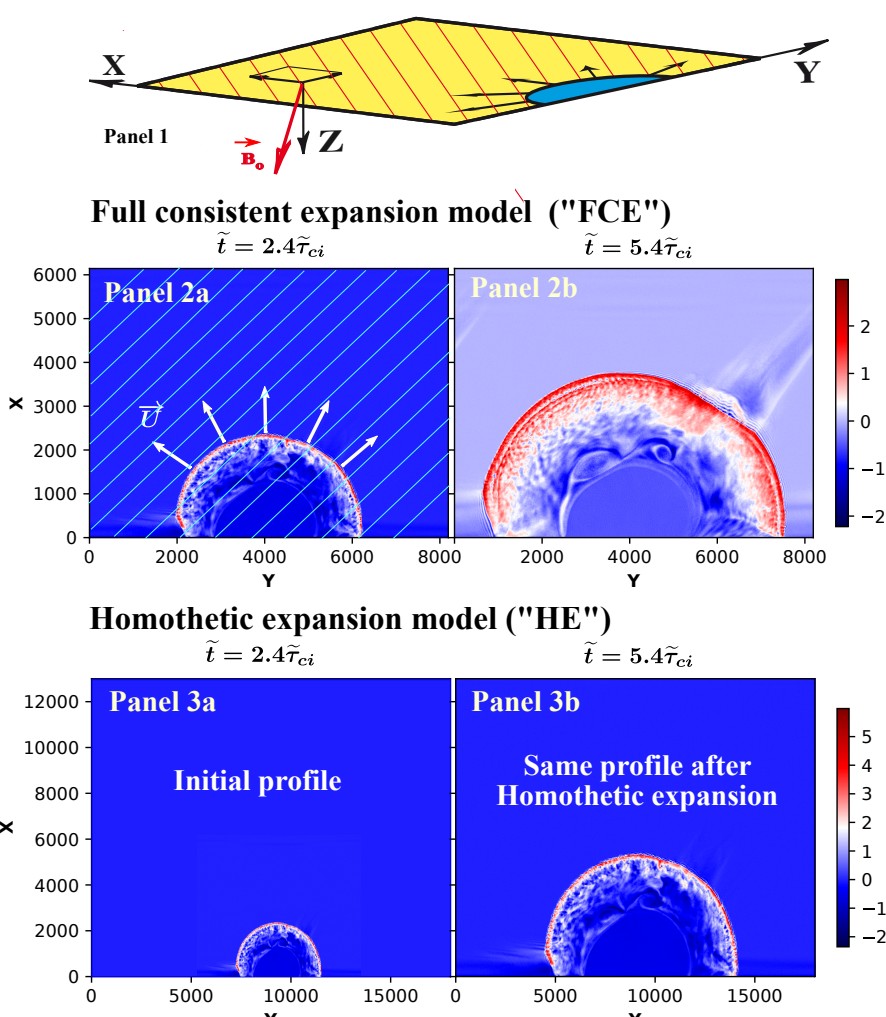

**Figure 1.** Panel 1 plots the simulation plane geometry; $\widetilde{B}_o$ magnetostatic field is mainly outside and directed downward from the plane. Panels 2a-b illustrate the evolution of the magnetic field $\widetilde{B}_{tz}$ in the fully consistent expansion model "*FCE*" (time dependent) respectively at $\widetilde{t}_{init} = 2.4\widetilde{\tau}_{ci}$ and $\widetilde{t}_{simul} = 5.4\widetilde{\tau}_{ci}$. In Panel2a, used as a reference, the vector velocity $\widetilde{U} = \widetilde{v}_{shock}$ has been superimposed to illustrate the shock front propagation (white thick arrows); the arrow length is not at the right scale. In addition, the projection of the $\widetilde{B}_o$ magnetic field lines has been reported (oblique white thin lines). The two Panels 3a-b illustrate one example of the curved magnetic field $\widetilde{B}_{tz}$ in the homothetic expansion model "*HE*" (time independent), where the shock profile is fixed in time but expands in space via an expanding factor proportional to the shock front velocity $v_{shock} * t$; this shock profile has been chosen at time $\widetilde{t}_{init} = 2.4\widetilde{\tau}_{ci}$ of the self-consistent simulation. From this time, the shock front dilates by a factor of $2.6$ compared to its initial shape.

In our configuration, the magnetostatic field is partially lying outside the simulation plane (see Savoini and Lembège (2001) for more details). Then, the simulation is limited to the whole quasi-perpendicular shock (i.e., for $45° \leq \theta_{Bn} \leq 90°$). We use a magnetic piston whose geometry is adapted to initiate a shock front with a curvature radius large enough as compared with the upstream ion Larmor gyroradius $\widetilde{\rho}_{ci}$ (all normalized quantities are indicated with a tilde "~", this the same normalization which is used in the previous self-consistent PIC simulations (Savoini and Lembège, 2001; Savoini et al., 2013; Savoini and Lembège, 2015). The curvature increases during the simulation. This configuration has two consequences: (i) first, ~~as $\theta_{Bn}$ decreases from 90°,~~ as the time increases and the shock front expands, its velocity slightly decreases, and so does the Alfvén Mach Number $M_A$ from $\approx 5$ to $\approx 3$, where the velocity is measured at $\theta_{Bn} = 90°$ used as a reference angle; (ii) the "time-of-flight" effects are self-consistently included. Indeed, this ballistic process is observed when the upstream magnetic field lines are convected by the incoming solar wind. In present simulations (based on upstream rest frame), the curved shock front expands and scans different $\theta_{Bn}$ values. As a result, backstreaming particles, collected at a given upstream location, come from different parts of the curved shock front depending on their respective velocity.

Initial plasma conditions are summarized as follows: light velocity $\widetilde{c} = 3$, temperature ratio between ion and electron population $T_{el}/T_{io} = 1.58$. A mass ratio $m_i/m_e = 84$ is used in order to save CPU time and the Alfvén velocity is $\widetilde{v}_A = 0.16$. The simulation plane size is $NCX = NCY = 8192 \approx 150\widetilde{\rho}_{ci}$ with the size of a grid-cell $\Delta_x = \Delta y \approx 1\widetilde{\rho}_{ce}$. The shock is in supercritical regime with a time averaged Alfvén Mach number $M_A \approx 4$ measured at $\theta_{Bn} = 90°$. In order to observe the early stage of the ion foreshock formation, the end time of the simulation is $\widetilde{t}_{simul} = 5.4\widetilde{\tau}_{ci}$ (where $\widetilde{\tau}_{ci}$ is the upstream ion gyroperiod), which is large enough to investigate the interaction of incoming ions with the shock front and the further formation of backstreaming ions.

## 2.2 Test particle simulations

In the present paper, we use all field components issued from the same previous PIC simulation as in Savoini and Lembège (2015); all components have been saved every $\Delta\widetilde{T} = \widetilde{\tau}_{ci}/20$. Test-particle simulations reveal to be a straightforward way to evaluate the action of different field components on the ion dynamics. Indeed, the feedback effects of particles on electromagnetic fields are excluded in test particle simulations, and one can modify or cancel some field components independently one from each other. This allows to identify their specific actions on particles and on the resulting ion reflection processes.

Figure 1 plots an example of the two configurations used hereafter in this paper. Panels 2a and 2b show the ***Fully Consistent Expansion model*** (hereafter named *FCE*) which corresponds to results where test ions interact with the $\overrightarrow{E}$ and $\overrightarrow{B}$ fields issued from the self-consistent simulation and where both spatial inhomogeneities and nonstationarities are fully included. If this configuration is easily to understand, the so-called particular approach named *"Homothetic Expansion" model* (hereafter named "*HE*") shows in Panel 3a-b is complementary. In this case, particles interact with propagating "fixed front profile" i.e., all-time profile variations are excluded; only spatial inhomogeneities of the shock front profile chosen at the selected time are included, as detailed in Section 4.

In the two configurations "*FCE*" and "*HE*", we inject test particles distributed within 10 individual sampling boxes located along the curved shock front (Figure 2). This procedure allows us to analyze the impact of the front curvature (local $\theta_{Bn}$) on

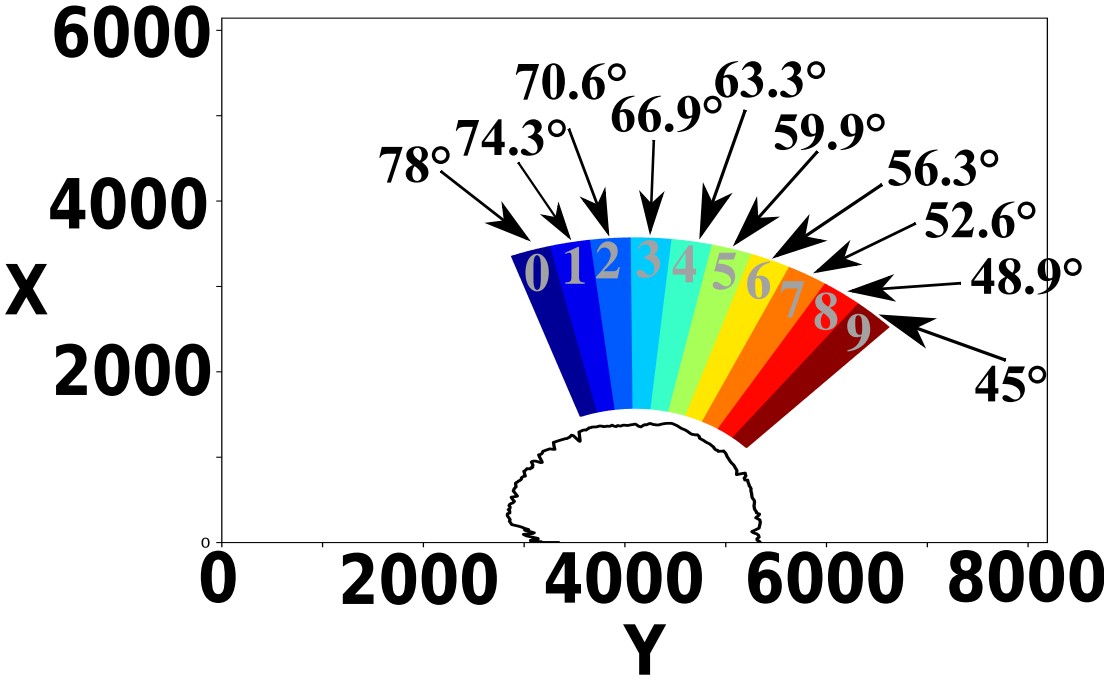

**Figure 2.** Initial location of the 10 sampling boxes (labelled from 0 to 9) which map the upstream ion foreshock region. All ions belonging to a given box are represented by the same color for statistical analysis only (sections 3 and 4). The $\theta_{Bn}$ propagation angle where each box is initially centered at time t= 0 is reported above the corresponding identification number of the box, but these colors will not be used anymore in this paper.

the formation of backstreaming ions. We follow a total of 1 million test particles. Then, each box has the same number of particles $N = 100000$ and are initialized as a Maxwellian distribution with a thermal velocity $v_{thi}$ which is the same as in the self-consistent simulation (Savoini and Lembège, 2015).

~~For reasons of clarity, we have associated a different color to each box and all ions belonging to a given box have the same color.~~

Let us point out that the use of finite size sampling boxes at different initial $\theta_{Bn}$ angles does only estimate the location where the particles hit the shock but does not provide an exact value for the local $\theta_{Bn}$ seen by the particles when these interact with the expanding shock front. Nevertheless, it reveals to be quite helpful when classifying the different types of particle interactions with the curved front. The sizes of all identical boxes are chosen so that (i) along the curved shock front, each box has an angular extension of $\approx 4^o$ which is small enough to scan the different orientations of $\theta_{Bn}$ and large enough for statistical constraints, and (ii) along the local shock normal, each box has a length large enough ($L_{size} = 2000\Delta x$) to ensure that most particles interact with the shock front during a noticeable time range (i.e., $D_T \approx 3\widetilde{\tau}_{ci}$).

Then, section 3 will present results obtained in the "*FCE*" model which is the usual configuration representing the time evolution of test particles with self-consistent shock front profile. The section 4 will introduce the more unusual "*HE*" expansion model (i.e. homothetic simulation approach).

## 3 Numerical results: the Fully Consistent Expansion Model "*FCE*"

For clarifying the presentation, we will split this section into two parts where we analyze respectively (i) the dynamics of the backstreaming ions (alias "*BI*") in the different boxes and their main features in terms of spatial and time evolution, and (ii) their behavior when some field components at the shock front are included/artificially excluded.

### 3.1 General features of the backstreaming ions

Figure 3 plots the spatial distribution of backstreaming ions density for the different sampling boxes (defined in Figure 2) at the end time of the simulation for the "*FCE*" model. Different information can be summarized as follows:

(i) The percentage of the backstreaming ions $BI_\%$ is obtained by computing the ratio of the backstreaming ions over the number of ions which have interacted with the shock front. This number increases when moving further into the foreshock (i.e. for decreasing $\theta_{Bn}$) from $BI_\% = 0.1$ for $N_{Box} = 0$ to $BI_\% \approx 14$ for $N_{Box} = 9$. This $\theta_{Bn}$ dependence is in agreement with previous experimental observations (Ipavich et al., 1981; Eastwood et al., 2005; Mazelle et al., 2005; Turner et al., 2014) and numerical simulations (Savoini and Lembège, 2015; Kempf et al., 2015).

(ii) The upstream edge of the ion foreshock (dashed line in Figure 3 for $N_{Box} = 10$) is not parallel to the IMF but is the result of the "time-of-flight" effect included in our simulations (Savoini et al., 2013). At the end of the simulation, this edge starts from the shock at the same critical angle so called $\theta_{io,fore} \approx 70^o$, as that found in our previous self-consistent simulations (Savoini et al., 2013; Savoini and Lembège, 2015).

(iii) We observe that the backstreaming ion density is not uniform along the shock normal but exhibits different maxima. Not only the spatial distribution is not the same for all boxes but is even not uniform within a given same box, i.e. backstreaming ions do not escape uniformly away from or along the shock front. For instance, boxes $N_{box} = 0 - 3$ evidence two distinct "spots" near the shock front indicated by black arrows. As $\theta_{Bn}$ decreases (i.e. $N_{box} = 5 - 9$) the right-hand "spot" disappears and backstreaming population increasingly aligns along the upstream magnetic field $B_o$. Accordingly, the width of the reflection area (i.e. the angular extension of the ion foreshock defined very near the shock front) shrinks from $\approx 50\rho_{ci}$ (for $N_{Box} = 0$) to $\approx 17\rho_{ci}$ (for $N_{Box} = 9$).

These two distinct "spots" may be explained by the different time histories of the backstreaming ions within the shock front as reported in Savoini and Lembège (2015). Actually, the interaction time strongly differs from one ion to another depending on its gyrating feature when it hits the shock front for the first time. Short and long interactions time can be defined depending on whether the reflection process is respectively associated to a short or long displacement of the ion along the shock front before escaping upstream. If the individual trajectory of the reflected ions has been already evidenced in Savoini and Lembège (2015), present test-particle simulations allow to generalize the results via a statistical approach versus their initial angular

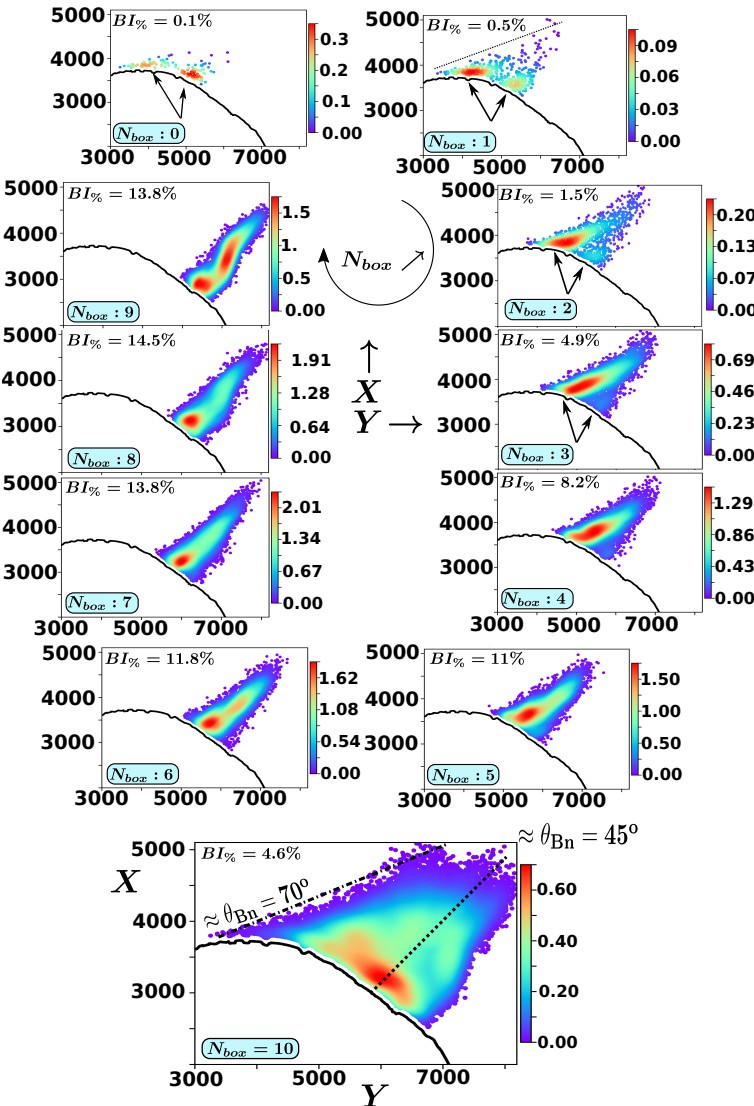

**Figure 3. "FCE"** configuration: Spatial distribution of the backstreaming particle density within the simulation $X - Y$ plane at the end of the simulation time ($\widetilde{t}_{end} \approx 5.4\widetilde{\tau}_{ci}$ where $\widetilde{\tau}_{ci}$ is the upstream ion cyclotron period). All boxes are plotted from $N_{box} = 0$ to 9 defined initially at $\theta_{Bn} \approx 90°$ and $\approx 45°$, respectively; the bottom panel $N_{box} = 10$ shows aggregate boxes where we have reported the edge of the ion foreshock (dashed line) and the angle $\theta_{Bn} = 45°$ (dotted line) for reference. Considering the small number of ions involved in the reflection process, we have used a gaussian interpolation which gives the relative density weight of each ion. Then, the color code (vertical bar) gives only an indication of the relative density amplitude. The location of the curved shock front is defined at the middle of the front ramp (thick black line) at the last time $\widetilde{t}_{end}$. Moreover, we have reported the space integrated percentage value $BI_\%$ of backstreaming ions within each corresponding box. In order to exclude the gyrating ions present near the front from the backstreaming population, we have eliminated ions being within a small area $\approx 2 - 3\rho_{ci}$ upstream of the shock front. For this reason, a very thin white area is visible along the curved front where no particle is present. For $N_{box} = 0 - 3$, the arrows point to the two spots (see the text for explanations).

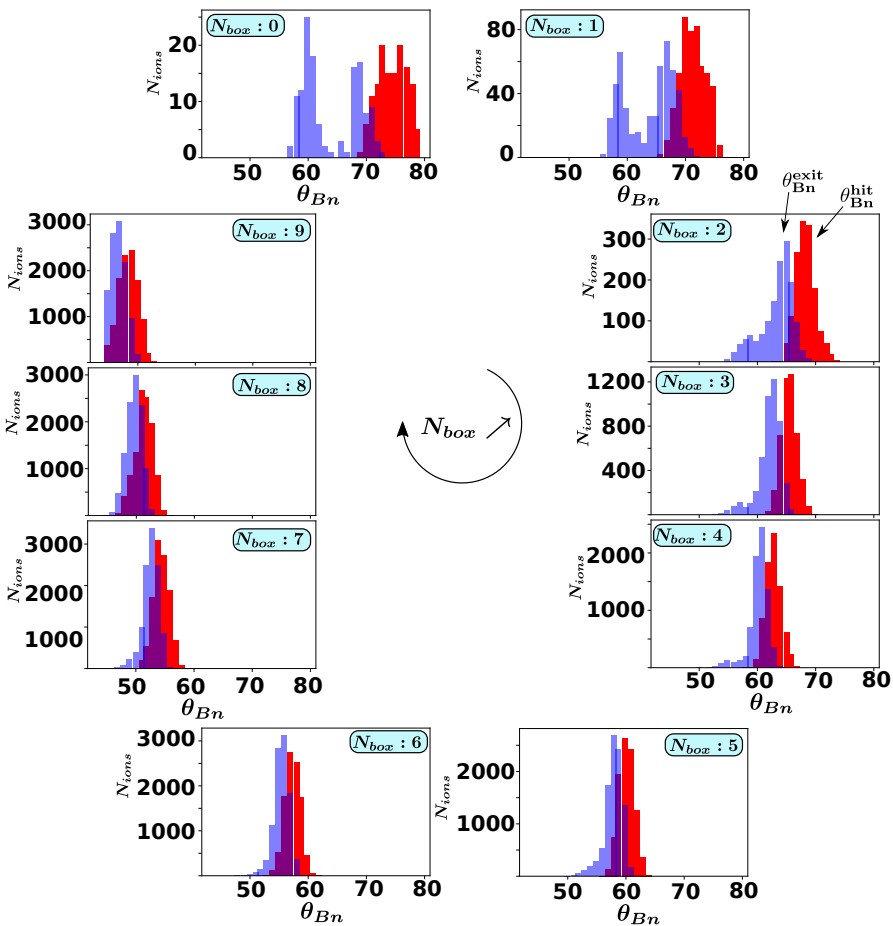

**Figure 4.** "FCE" configuration: Plots of the ion distribution functions for each box $N_{box} = 0 - 9$ versus the local $\theta_{Bn}$ angle computed when ions hit for the first time the shock front (red distribution function of the so called $\theta_{Bn}^{hit}$) and when these leave it and escape upstream (blue distribution function versus the so-called $\theta_{Bn}^{exit}$). The angles $\theta_{Bn}^{hit}$ and $\theta_{Bn}^{exit}$ have been reported in $N_{box} = 2$ for reference.

locations (i.e. the $N_{box}$ number). Figure 4 plots the distribution of $\theta_{Bn}$ angle seen by the particles when these hit for the first

time the shock front (hereafter named $\theta_{Bn}^{hit}$ in red color) and when they finally exit the shock front to escape upstream (hereafter named $\theta_{Bn}^{exit}$ in blue color). These statistical results are obtained by computing these angles for each particle. As a consequence, angle values are computed neither at the same time, neither at the same location along the curved front (even if they are initially located in the same box). In other words, each particle sees different local shock front profiles in terms of spatial inhomogeneity and time nonstationarity of the shock front.

First, let us note that the averaged value of $\theta_{Bn}^{hit}$ corresponds mainly to the initial location of the box, and therefore, the important feature is not the angle values themselves but rather, the difference between the averaged values of $\theta_{Bn}^{hit}$ and $\theta_{Bn}^{exit}$

when ions hit and leave the shock front respectively. For this reason, we will use the angular range of particles interaction with the front defined by $\Delta_{int}\theta_{Bn} = \theta_{Bn}^{exit} - \theta_{Bn}^{hit}$. Obviously, $\theta_{Bn}^{hit}$ decreases as $N_{box}$ increases until approaching the limit of the quasi-perpendicular domain of propagation i.e. $\theta_{Bn} = 45^o$ for $N_{Box} = 9$.

Second, $\theta_{Bn}^{hit}$ reveals to be a good reference entity to be compared with the escaping angle $\theta_{Bn}^{exit}$. Indeed, distribution functions of $\theta_{Bn}^{exit}$ strongly differ according to the concerned box. For $N_{Box} = 0-2$, two (blue) peaks occur: one for high $\theta_{Bn}^{exit}$ (for which $\Delta_{int}\theta_{Bn} \approx 4-5°$), the other for lower $\theta_{Bn}^{exit}$ ($\Delta_{int}\theta_{Bn} \approx 15°$). In terms of time trajectory, the presence of these two peaks suggests that some ions have spent different interaction times (subscript "$_{int}$") within the shock front. Some escape after a short interaction time (i.e. small $\Delta_{int}\theta_{Bn}$) while others escape after a long interaction time (i.e. large $\Delta_{int}\theta_{Bn}$), where the

terms short and long refer to a small and large drift along the shock front as already analyzed in Savoini and Lembège (2015). In other words, small drift refers to one bounce whereas large drift refers to multi-bounces process along the shock front.

Moreover, as $N_{box}$ increases (i.e. $N_{box} \geq 3$), the lower $\theta_{Bn}^{exit}$ distribution (i.e. correspondingly the largest $\Delta\theta_{Bn}$) decreases rapidly in amplitude and disappears from $N_{Box} = 6$ (i.e. $\theta_{Bn}^{hit} \leq 56^o$). Simultaneously, the other peak (i.e. correspondingly the smaller $\Delta_{int}\theta_{Bn}$) becomes dominant for all higher order boxes meaning that less and less ions are associated to large drifts

along the shock front.

Third, in order to complete information deduced from Figure 4, Figure 5 plots the number of reflected ions versus the time spent within the shock front. This interaction time $\Delta\widetilde{T}_{int}$ is defined as the time difference between the time associated to $\theta_{Bn}^{exit}$ and to $\theta_{Bn}^{hit}$. Different main maxima of backstreaming ions density are evidenced namely $f_1$, $f_2$; a third maximum $f_3$ can be also observed for boxes $N_{box} = 0-4$ but its amplitude is too weak to be relevant in this discussion. One important feature is that $f_1$

and $f_2$ appear in all boxes and are independent of the box number. More precisely, $f_1$ appears about $\Delta\widetilde{T}_{int} \approx 0.25\widetilde{\tau}_{ci} \approx \widetilde{\tau}_{ci}^{shock}$ while $f_2$ is observed at $\Delta\widetilde{T}_{int} \approx 1\widetilde{\tau}_{ci} \approx 4\widetilde{\tau}_{ci}^{shock}$ where $\widetilde{\tau}_{ci}^{shock}$ is the local gyroperiod estimated within the shock front (at the middle of the ramp). This indicates that the reflection process is not uniform in time but leads to the formation of ion "bursts" associated to the shock dynamics even if the number of ions which spend several gyroperiods $\widetilde{\tau}_{ci}^{shock}$ (i.e. $\approx 4$ bounces) within the shock front is rapidly negligible. In addition, for $N_{Box} = 0-2$, $f_1$ and $f_2$ have a similar amplitude which is not the case for

$N_{Box} = 4-9$. In fact, a close look of $f_1$ and $f_2$ shows that $f_2$ does not decrease in magnitude but rather the amplitude of $f_1$ drastically increases from 10 ($N_{box} = 0$) to 2500 ($N_{box} = 9$). Then, the $f_2$ population is always present but becomes negligible for lower $\theta_{Bn}$ as compared with $f_1$; this explains why we do not observe two distinct "spots" for $N_{box} = 4-9$.

One helpful aspect of the test particle approach is to include or exclude some electromagnetic field components in order to analyze their impact on the particles dynamics. Indeed, it is clear that some electric field component (i.e. $\overrightarrow{E}_l \times \overrightarrow{B}$ drift)

as well as strong magnetic gradients drift (i.e. $\propto -\overrightarrow{\nabla}B \times \overrightarrow{B}$ drift) can be a prerequisite for a large drift along the front (i.e. $\Delta_{int}\theta_{Bn} \approx 15°$) whereas it could be unnecessary for the other case $\Delta_{int}\theta_{Bn} \approx 4-5°$. Unfortunately, the shock front magnetic gradient can not be cancelled without the shock itself, then, we will focus our study by including or not the electric components which will shed new light on the origin of backstreaming ions filling the foreshock.

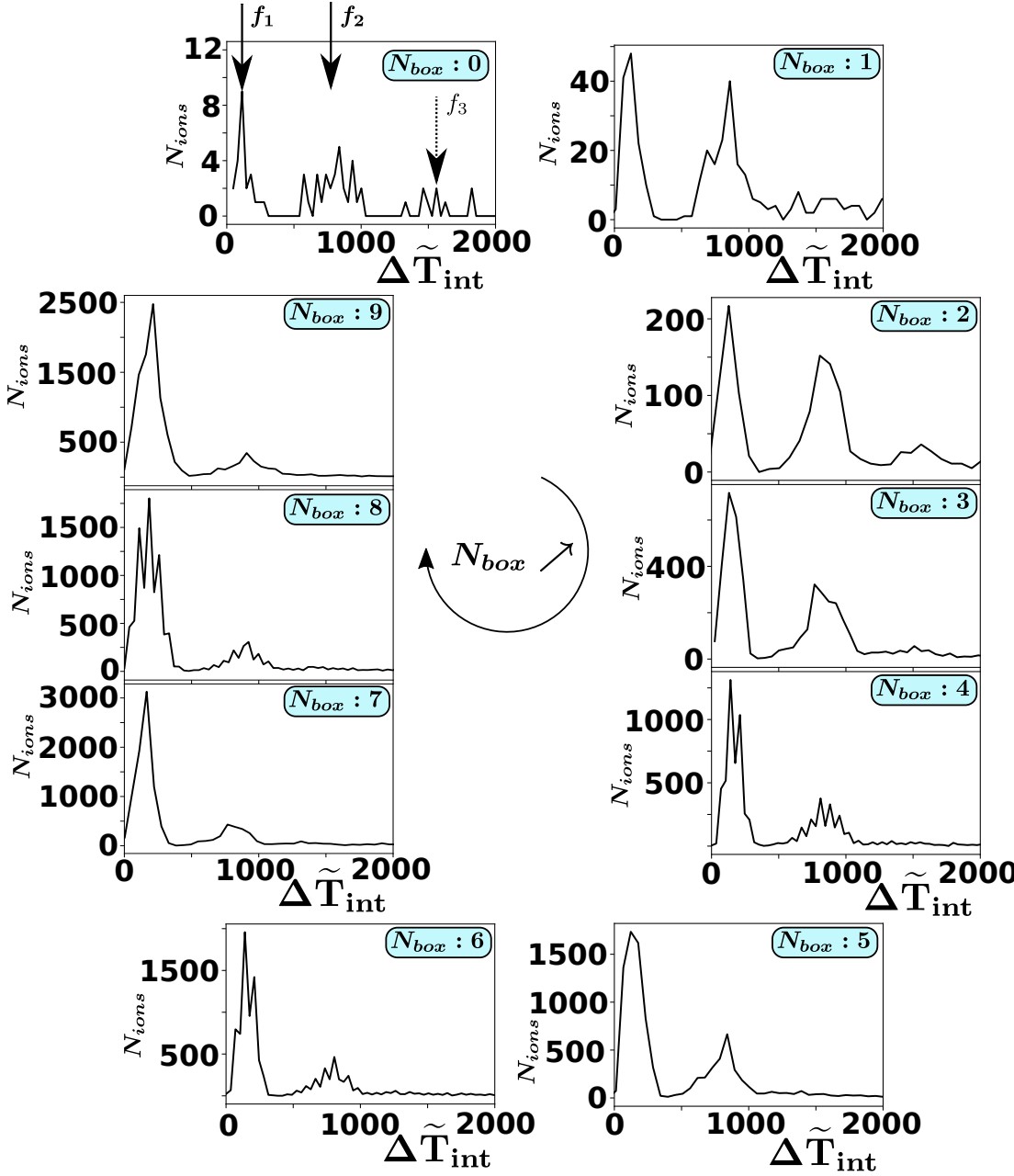

**Figure 5. "FCE"** configuration: Plots of the ion distribution function (for each box $N_{box} = 0 - 9$) versus the interaction time range $\Delta \widetilde{T}_{int}$ spent by each particle within the shock front. As shown, this interaction time range is not continuous but evidences distinct "bursts" of reflected ions (hereafter named $f_1$ and $f_2$), respectively defined at $\Delta \widetilde{T}_{int} \approx 250 \approx 0.25 \widetilde{\tau}_{ci}$ and $\Delta \widetilde{T}_{int} \approx 950 \approx 1 \widetilde{\tau}_{ci}$, where $\widetilde{\tau}_{ci}$ is the upstream cyclotronic period. A third "burst" $f_3$ of reflected ions can be identified around $\Delta \widetilde{T}_{int} \approx 1500 \approx 1.5 \widetilde{\tau}_{ci}$ for $N_{Box} = 0 - 2$, but can be neglected as compared with the others.

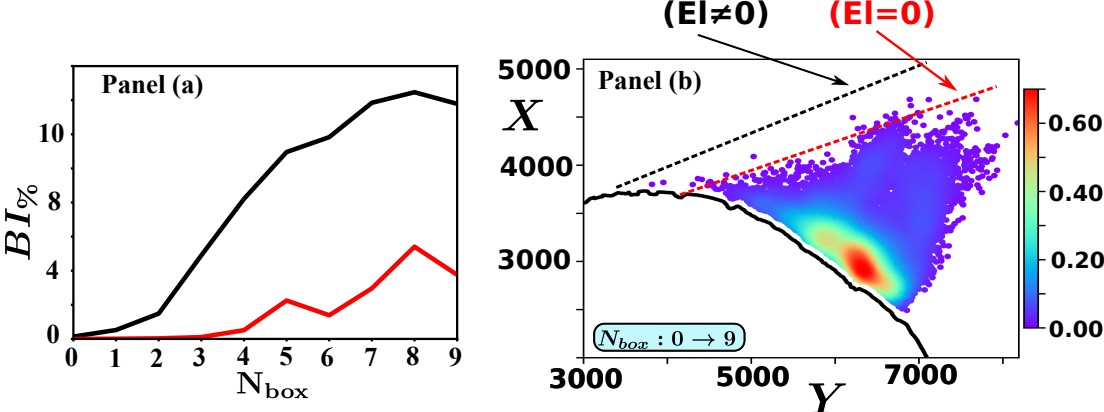

**Figure 6. "FCE"** configuration: Characteristics of the ion foreshock with and without the electrostatic field component $\widetilde{E}_l$ (i.e. $\widetilde{E}_{lx} = \widetilde{E}_{ly} = 0$). Panel (a) shows the percentage of the backstreaming ions $BI_\%$ versus the box number. Black and red straight dashed lines are defined for $\widetilde{E}_l \neq 0$ and $\widetilde{E}_l = 0$, respectively. Panel (b) shows the density of the backstreaming particles in the same format as Figure 3 but when $\widetilde{E}_l = 0$. Only the view which aggregates all boxes (i.e. $N_{box} = 10$) is shown in order to evidence the location of the edge of the ion foreshock (dotted line) in each case (in black when $\widetilde{E}_l \neq 0$ and in red when $\widetilde{E}_l = 0$, respectively).

## 3.2 Impact of electric field components

Savoini and Lembège (2015) have analyzed the impact of $\overrightarrow{E} \times \overrightarrow{B}$ drift velocity on the dynamics of backstreaming ions (Gurgiolo et al., 1983) and, more particularly, as a source of "FAB" and/or "GPB" populations. This study has shown that the origin of both populations can be easily explained in terms of $\overrightarrow{E} \times \overrightarrow{B}$ drift associated or not to a diffusion in the velocity space, but was not able to explain the details of the reflection mechanism itself. Then, herein, we will focus on the role of electrostatic field component $\widetilde{E}_l$ built up within the shock front (i.e. space charge effects). This longitudinal component, defined along the

normal to the shock front, can be associated to the electrostatic potential wall responsible of some reflected ions. In the case of a constant shock profile in time with a planar geometry, this reflection does conserve the energy since the potential is the same before and after the reflection, and the total work of the electric force is cancelled. Nevertheless, in more realistic conditions, this scenario is not valid anymore for ions which drift along the shock front and suffer both time and space electric and magnetic field variations. Then, in the following sections, we will use preferentially the field $E_l$ rather than the potential $\Phi$.

~~Moreover, the convective electric field component $E_t$ can not be canceled easily since it is induced by the propagating shock wave in the Solar wind reference frame.~~

Figure 6a shows the percentage $BI_\%$ of backstreaming ions versus the box number where the black and red curves are defined for $\widetilde{E}_l \neq 0$ and $\widetilde{E}_l = 0$ respectively. The impact of the $\widetilde{E}_l$ field (i.e. the potential wall) on the reflection process is clearly apparent for all $\theta_{Bn}$ values (namely for each $N_{box}$ number). In particular, the percentage $BI_\%$ strongly decreases as

$\widetilde{E}_l = 0$, which illustrates the dominant role of $\widetilde{E}_l$ field ~~in particular~~ what ever the box is. This is especially true, for lower

box number $N_{Box} = 0 - 3$ (i.e. high $\theta_{Bn}$ approaches $90^o$) where very few backstreaming ions are observed. ~~This point is not surprising if one reminds that for $N_{Box} = 0$ (i.e. largest $\theta_{Bn}$) escaping ions have to be accelerated to higher parallel velocity as reviewed in Burgess et al. (2012) and our simulations reveal that the electric component $\widetilde{E}_{t\parallel}$ is a good candidate to accelerate ions in the parallel direction.~~ This point is not surprising if one reminds that this electrostatic field decelerates the incoming ions (i.e. accelerates ions in present reference frame) and contributes to the reflection process. In other words, for $N_{Box} = 0$ (i.e. largest $\theta_{Bn}$) escaping ions have to be accelerated to higher parallel velocity as reviewed in Burgess et al. (2012). Let us stress that Figure 6a exhibits a clear change in the slope of $BI_\%$ increase at the box $N_{Box} = 2$ centered around $\theta_{Bn} = 70°$. Herein, we will consider this value as the reference angle identifying the starting location of the ion foreshock edge attached to the shock front. This value is in reasonable agreement with the value ($\theta_{io,fore} \approx 66^o$) found approximately in the previous self-consistent PIC simulations Savoini and Lembège (2015).

Another consequence is illustrated in Figure 6b, which shows that the edge of the ion foreshock is shifted due to the lack of reflected ions and starts around $\theta_{Bn} \approx 55°$. Clearly, the contribution of the electric field is important for ions which populate the edge of the foreshock and need to escape at high $\theta_{Bn}$.

Another way to observe the strong impact of $\widetilde{E}_l$ on the dynamics of reflected ions is illustrated in Figure 7 which shows the two $\theta_{Bn}^{hit}$ and $\theta_{Bn}^{exit}$ distributions in the same format as that of Figure 4. The number of backstreaming ions decreases drastically for all boxes, and is zero for $N_{box} = 0$. Furthermore, the density is not uniform for all boxes and appears to be much more important for $N_{Box} = 5 - 9$ than for $N_{Box} = 1 - 4$. The $\theta_{Bn}^{exit}$ distribution is strongly modified and a comparison between Figures 4 and 7 can be summarized as follows:

1. ~~The boxes $N_{Box} = 1 - 2$ evidence a total absence of reflected ions having a small range $\Delta_{int}\theta_{Bn}$ and only the $\theta_{Bn}^{exit}$ distribution around $60°$ persists. This result shows that $\widetilde{E}_l$ plays a key role in the formation of some backstreaming ions and more specifically, for the ions suffering a "one bounce" reflection whereas the ions suffering a large drift are mainly accelerated along the shock front by the $E_t$ induced electric field always included in the simulation due to the propagating shock front into the Solar wind.~~

   The boxes $N_{Box} = 1 - 2$ evidence a total absence of reflected ions having a small range $\Delta_{int}\theta_{Bn}$ and only the $\theta_{Bn}^{exit}$ distribution around $60°$ persists. This result shows that $\widetilde{E}_l$ field plays a key role more specifically on backstreaming ions suffering a "one bounce" reflection near the edge of the ion foreshock (i.e. $\Delta_{int}\theta_{Bn} \approx 4 - 5°$).

2. The $\theta_{Bn}^{exit}$ distribution in both cases ($\widetilde{E}_l \neq 0$ in Figure 4 and $\widetilde{E}_l = 0$ in Figure 7) is roughly similar in corresponding boxes whatever $N_{Box} \geq 6$. This can be interpreted either as the ions have been enough accelerated during their reflection at the shock front or as they need a lower parallel velocity to escape upstream. As a consequence, the $\overrightarrow{E}_l$ component is not anymore mandatory and the mirror magnetic mechanism ~~or Fermi reflection~~ at the shock front can be invoked as the only reflection process.

In summary, the comparison between figures 4 and 7 evidences that $\widetilde{E}_l$ components are essential in the ion reflection for high $\theta_{Bn}$ angle ($> 56°$, i.e. $N_{Box} = 6 - 9$) where ions need strong acceleration process but play a less important role at lower $\theta_{Bn}$

angles. Conversely, for $N_{Box} = 6-9$, the reflection process takes place with a very small $\Delta_{int}\theta_{Bn}$ with or without the electric field. In other words, the large shock drift invoked for $\theta_{Bn} \leq 56°$ seems to be mainly supported by the convective electric field $\widetilde{E}_t$ components present at the shock front.

Similarly, the "one bounce" reflection always occurs even in absence of $\widetilde{E}_l$ field (i.e. in absence of shock front potential wall) and then, can be associated to a magnetic reflection ~~Fermi type one acceleration~~ process which seems to be very efficient especially at lower $\theta_{Bn}$. This "one bounce" reflection (i.e. $f_1$) corresponds essentially to a short interaction time as illustrated in Figure 5 ($\Delta\widetilde{T}_{int} \approx 1\widetilde{\tau}_{ci}^{shock}$). Then, the ion energy gain is essentially due to a Fermi type acceleration.

### 3.3 Impact of the shock front nonstationarity

Previous studies have largely evidenced that a quasi-perpendicular shock front can be intrinsically non stationary due to different mechanisms (for a review see Lembege et al. (2004); Marcowith et al. (2016)). Then, it is important to analyze the impact of such non stationarity on the temporal ion foreshock dynamics. As a first step, we plot in Figure 8 the time evolution of the backstreaming ions percentage $BI_\%$ as these leave the front and escape into the upstream region, where $BI_\%$ is the instantaneous rate computed during a short time range $\Delta\widetilde{T} = \widetilde{\tau}_{ci}/20$. The time $\widetilde{T}_{init} = 1248$ is the initial time when test-particles are launched into the time-dependent simulation.

(i) Results of Figure 8 are obtained as the $\widetilde{E}_l$ field components are included (black curve) and artificially excluded (red curve). One retrieves that the percentage $BI_\%$ strongly decreases as $\widetilde{E}_l$ components are excluded and that the impact of $\widetilde{E}_l$ field is emphasized for lower $N_{box}$. In other words, the backstreaming ions mainly appear for higher $N_{box} > 5$ (i.e. for lower $\theta_{Bn}$) even in absence of $\widetilde{E}_l$ field.

ii) The different results may be classified into two groups: (i) a first one concerns boxes $N_{Box} = 0-4$ showing a "slow" increase (almost monotonic) of the reflection rate and (ii) a second group $N_{Box} = 5-9$ which evidences a "steep" increase followed by a "flat-top" shape around $BI_\% \approx 1$, even if it increases slightly with $N_{Box}$. At the end of the simulation, the strong decrease of $BI_\%$ observed for all boxes corresponds to the time when all ions of the different boxes have been swapped by the propagating shock front and then, no more ions are backstreaming.

For the first group $N_{Box} = 0-4$, a delay is observed in the formation of backstreaming ions between the different boxes although test-particles are initially evenly distributed in the whole boxes ~~For the first group, even if all test-particles are initially released at the same distance from the shock front, a delay is observed in the formation of backstreaming ions between the different boxes.~~ For $N_{Box} = 0$, backstreaming ions appear around $\widetilde{T} \approx 4000$ (i.e. $\approx 2.6\widetilde{\tau}_{ci}$) from the initial release time $\widetilde{T}_{init}$, whereas this time delay decreases to $\widetilde{T} \approx 770$ (i.e. $\widetilde{T} \approx 0.5\widetilde{\tau}_{ci}$) as $N_{Box}$ increases. This illustrates the larger time delay of ions having interacted with the front to escape upstream at high $\theta_{Bn}$. For $N_{Box} = 0-2$, ions have to stay longer within the shock front to finally escape at lower $\theta_{Bn}^{exit}$ (Figure 7) which is illustrated by the increase of $BI_\%$ as the time evolves (Figure 8). The second group ($N_{Box} = 5-9$) concerns boxes which are already at lower angle $\theta_{Bn}$ with easier escaping conditions. In this case, ions are reflected continously with some time variation in $BI_\%$.

iii) Another interesting feature is the presence of different modulations which are superimposed to a time averaged reflection ion rate (blue dashed line), especially for the boxes $N_{box} = 1-5$ and the boxes $N_{box} = 7-8$. Nearly all boxes exhibit these

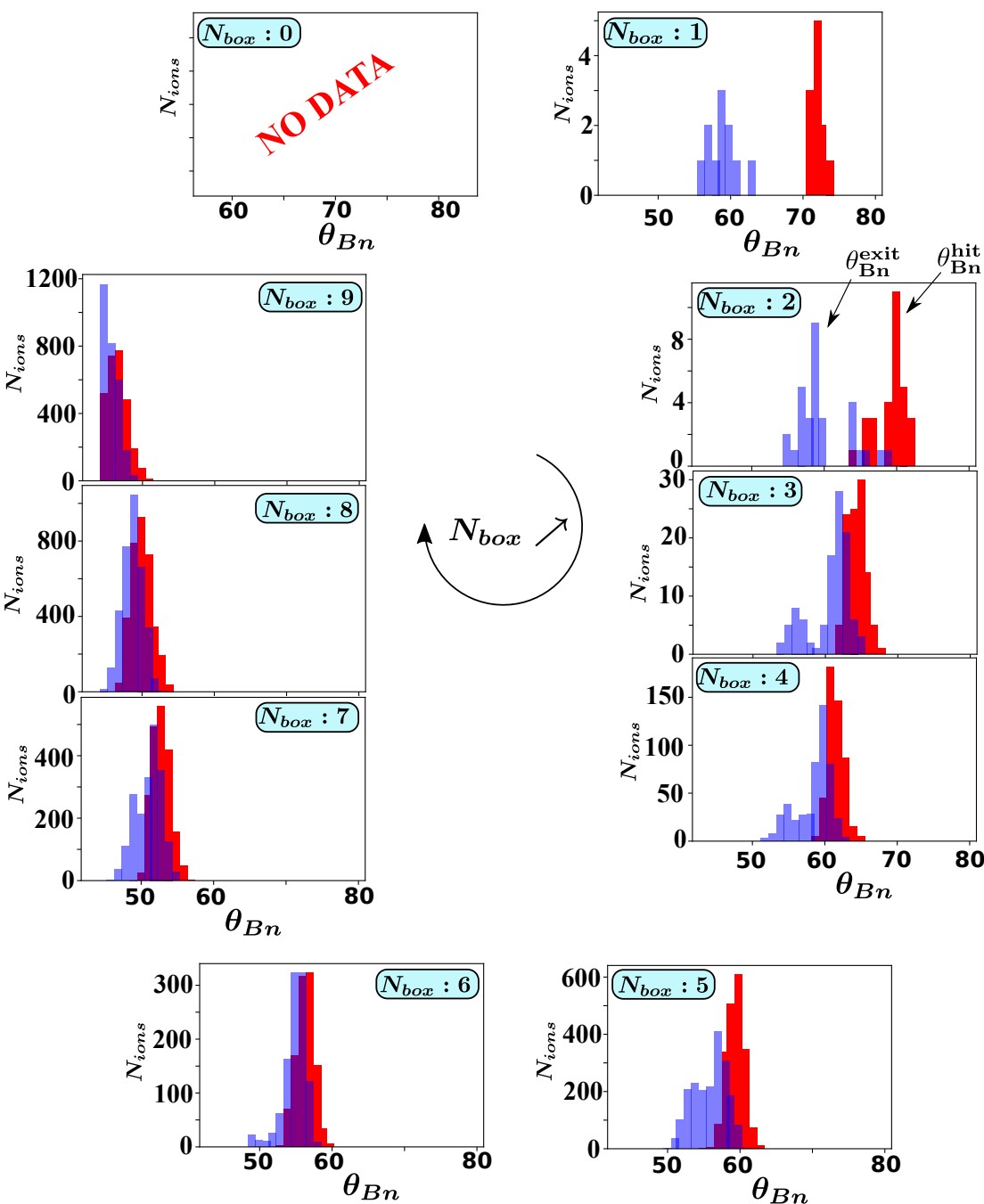

**Figure 7. "FCE"** configuration: Same plots as for Figure 4 but when electrostatic field components $E_{lx}$ and $E_{ly}$ are artificially excluded ($E_l = 0$). The angles $\theta_{Bn}^{hit}$ and $\theta_{Bn}^{exit}$ have been reported in $N_{box} = 2$ for reference.

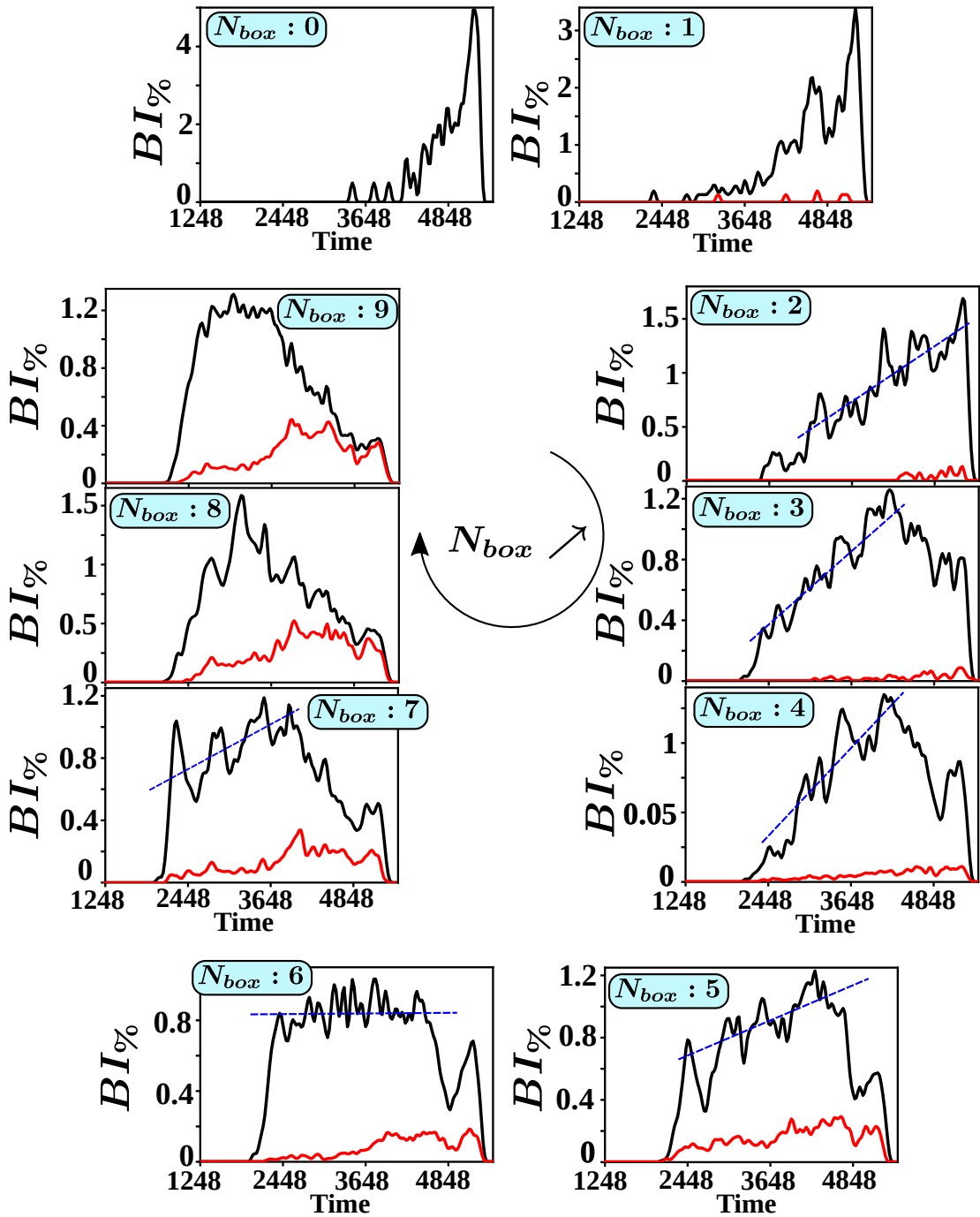

**Figure 8. "FCE"** configuration: Time history of the reflected ion percentage $BI_\%$; each value is computed during a short integrated time interval $\Delta\widetilde{T} = \widetilde{\tau}_{ci}/20$ for the different boxes. Within this interval, only the newly reflected ions are memorized. As for the Figure 6, black and red lines correspond to the case where electric field is included ($\widetilde{E}_l \neq 0$) and excluded ($\widetilde{E}_l = 0$), respectively. We have used the same normalization in order to compare both cases. The blue dotted straight lines are not obtained from a data linear approximation but only indicate the mean values of the time modulations.)

modulations which represent about $40\%$ of the averaged $BI_\%$, although these modulations amplitude varies versus time and $N_{box}$. These evidence a nonstationary ion escaping rate. These modulations almost disappear in the case $\widetilde{E}_l = 0$ (red line). This illustrates that the presence of the $\widetilde{E}_l$ field component is a key ingredient in the formation of these modulations since this electric field component is also involved in the shock front self-reformation as described in previous works (Lembege and Savoini, 1992; Scholer et al., 2003; Matsukiyo and Scholer, 2006).

This result confirms the importance of the electrostatic field component at the shock front in the reflection processes of the backstreaming ions, and most importantly, on the ion foreshock non stationarity behavior as described in a previous paper (Savoini and Lembège, 2015). Nevertheless, it is quite difficult to establish a one-to-one correspondence between these modulations and the non stationarity of the shock front because during the sampling time interval $\Delta\widetilde{T} = \widetilde{\tau}_{ci}/20$ the front non stationarity and the "time-of-flight" effects have mixed ions coming from either different times and/or different $\theta_{Bn}^{exit}$ regions (even if they are in the same box). So this ~~fully~~ self-consistent approach is not totally adapted to resolve this question. A complementary approach is necessary based on simplified simulations with fixed shock front profiles in expansion (nonstationary effects are excluded). This motivates the "**H**omothetic **E**xpansion" model ("*HE*") described in the next section.

## 4 Numerical results: the Homothetic Expansion ("*HE*") model

### 4.1 Descriptions of the "*HE*" model

Let use remind that all simulations are made in the Solar Wind reference frame (i.e., the curved shock expands into the "upstream region" where the Solar Wind is at rest). As a consequence, if one follows test particles within this configuration, we have to mimic this behavior. In order to proceed, we apply a homothetic transformation (homogeneous dilatation in all directions) with an expansion factor deduced from the shock front velocity determined at selected times as illustrated in Panels $3a - b$ of Figure 1. Special attention has been taken in the determination of this homothetic factor $\lambda = v_{shock} * t$ ~~in order to include an exact induced/convective electric field $\widetilde{E}_t$ (due to the relative motion between the Solar Wind and the shock front).~~ since the shock front velocity $v_{shock}$ at a given time must fit with the corresponding value issued from the PIC simulations (Savoini and Lembège, 2015). With this information, we are able to expand the shock front through a cubic interpolation as it propagates with $v_{shock}$ in an "expanding" simulation plane (i.e. the grid-cell stay constant $\widetilde{\Delta}_x = \widetilde{\Delta}_y = cte$ but the number of the grid-cell increases accordingly). In other words, all points of electromagnetic fields at the shock front follow the relation $\overrightarrow{OM} \longmapsto (v_{shock} * t)\overrightarrow{OM}$ where $v_{shock}$ is the value of the shock velocity as computed from our self-consistent 2D PIC simulations at the selected time and $\overrightarrow{OM}$ is the vector between the initial location of the shock front (i.e. the point $O$) and any point of the field array (i.e. the point $M$). At this stage, we have to point out that the velocity $v_{shock}$ remains artificially constant during the whole simulation which is not the case for the "*FCE*" model where $v_{shock}$ slightly decreases. Then, the same procedure is repeated for another selected time, so that one can analyze the impact of different shock front inhomogeneities and curvature on ion dynamics; let us note that time of flight effects are always included. Each front profile is analyzed within a same simulation time range $\approx 3\widetilde{\tau}_{ci}$. In summary, similar simulations are performed for 174 different times

320 in order to simulate all the different shock profiles provided by the 2D PIC self-consistent simulation from $\widetilde{t}_{init} = 1.2\widetilde{\tau}_{ci}$ to $\widetilde{t}_{simul} = 5.4\widetilde{\tau}_{ci}$.

## 4.2 General features of the backstreaming ions

Figure 9 has been achieved by performing 100 independent simulations (i.e. we take only the first 100 simulations so that all test-particles hit the propagating shock front). ~~For each simulation, we have followed a propagating "homothetic" shock over~~
325 ~~a same time range ($\approx 3\widetilde{\tau}_{ci}$).~~ During this range, the shock front is "forced" to expand (see section 4.1). For example, the chosen time $\widetilde{T} = 4456 \approx 4.2\widetilde{\tau}_{ci}$ on the abscissa axis corresponds to one particular shock front profile (i.e. including all electromagnetic field components issued from the previous self-consistent PIC simulations), from which we have measured the instantaneous shock front velocity and that we follow during the time range covering $\approx 3\widetilde{\tau}_{ci}$.

~~We have selected the 100 last shock profiles of the self-consistent simulation, since these are characteristic of a well de-~~
330 ~~velopped curved shock with a large curvature radius.~~ The purpose is to determine (i) whether some shock profiles are more appropriate than others for the formation of backstreaming ions and (ii) if yes, whether better reflection takes place for some particular angular range of $\theta_{Bn}$.

The comparison of Figures 8 and 9 provides the following information. The maximum $BI_\%$ value is much higher for the "*HE*" model than for the previous "*FCE*" model for each corresponding box. For example, for $N_{Box} = 9$, $BI_\%^{max} \approx 1.2$ in the
335 *FCE* simulations as compared with $BI_\%^{max} \approx 15$ in the *HE* configuration. In fact, one has to remind that for *FCE* model the $BI_\%^{max}$ value represents an instantaneous reflection rate versus the shock front evolution, whereas this rate is a time integrated value for the *HE* model. Indeed, in this model, the shock front profile stays the same during the whole simulation and then, if this profile allows the reflection of some incident ions, they will be reflected continuously leading to a high $BI_\%$ number.

Obviously, the main information is not the $BI_\%$ value itself but rather its evolution versus time and for different shock
profiles. Other main results issued from Figure 9 may be summarized as follows:

1. The $N_{Box} = 0$ box evidences almost no reflection for the majority of the shock front profiles which indicates that nonstationarity effects present in the *FCE* configuration (i.e. Figure 3) are needed for feeding backstreaming ions along the edge of the foreshock.

2. Boxes $N_{Box} = 1 - 8$ show clearly some strong modulations in the percentage $BI_\%$ versus the shock profile of concern
which correspond to a quasi-periodic bursty emission of backstreaming ions. The $BI_\%$ rate reaches periodically a maximum value followed by a minimum around 0. The corresponding time period $\Delta\widetilde{T}_{max}$ is about $\approx 460 = 0.5\widetilde{\tau}_{ci}$ (between two successive maxima). The temporal width of each maximum is about $\Delta\widetilde{T}_{range} \approx 256 = 0.25\widetilde{\tau}_{ci}$. These modulations mean that conditions for the formation of backstreaming particles are not continuous but correspond to some specific shock front profiles. In addition, these modulations appear synchronized in time for the different boxes $1 - 7$ which im-
plies that the local reflection conditions are not strongly dependent of $\theta_{Bn}$ angle but rather depend on the shock profile at certain times.

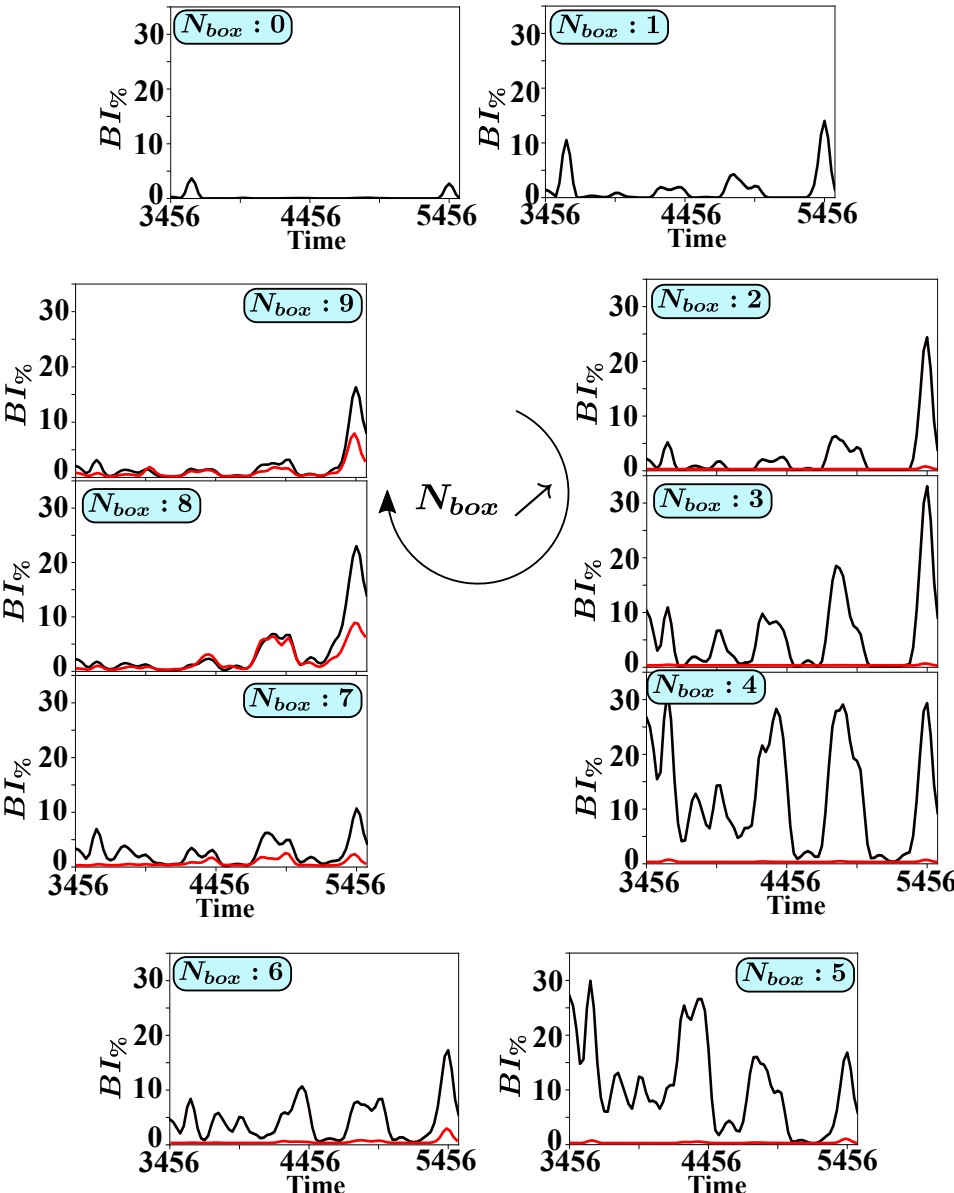

**Figure 9. "HE"** configuration: Percentage $BI_\%$ of backstreaming ions measured at the end of each simulation where each time corresponds to a given fixed shock front. For each shock profile in homothetic expansion, the simulation covers $3\widetilde{\tau}_{ci}$ allowing to obtain a well developed ion foreshock, and $BI_\%$ represents the ratio of the backstreaming ions over the total number of upstream ions which are released at the beginning of the simulation within a given box. As in Figure 7, black and red lines correspond to the case when $E_l$ field is included and artificially excluded, respectively. The concerned shock profiles are chosen only at late times of the full PIC simulation (from $\widetilde{T} = 3456$ to 5474) where the curvature radius of the shock front is relatively large ($\widetilde{R} > 70\widetilde{\rho}_{ci}$).

3. In contrast, the boxes $N_{Box} = 8 - 9$ evidence also the same kind of modulations but with greatly reduced amplitude; these are even nonexistent between $\widetilde{T} \approx 3456$ and $4456$ which indicates a low sensitivity to the shock front profile when approaching $\theta_{Bn} = 45°$.

4. Similarly, the maximum values $BI_\%^{max}$ (black curve) change drastically with box numbers: from small amplitudes $BI_\% \leq 6$ for $N_{Box} = 1 - 2$ to very high values $BI_\% \approx 30$ for $N_{Box} = 3 - 5$ before decreasing again for $N_{Box} = 6 - 9$. These variations may be understood by taking into account the reflection processes present at these different $\theta_{Bn}$ and more specifically, in regards to the electric field components. For boxes $N_{box} = 0 - 2$ reflection is almost impossible without the $\widetilde{E}_l$ component (i.e. electric potential wall). As $\theta_{Bn}$ decreases (for $N_{box} = 3 - 5$), the reflection becomes easier and both magnetic and electric field contribute to the percentage of reflected ions. Finally, for the last boxes $N_{box} = 6 - 9$, the peak amplitude decreases but the contribution of $\widetilde{E}_l$ becomes less important in the reflection process as evidenced by comparing both black ($\widetilde{E}_l \neq 0$) and red ($\widetilde{E}_l = 0$) curves. Instead, another process, essentially driven by magnetic field (mirror reflection) ~~(like Fermi type)~~ contributes more since the peak amplitude of the red curve increases progressively as $N_{Box}$ increases from 6 to 9.

5. Only the last "peak" around $\widetilde{T} \approx 5456$ has a different behavior in comparison with others. In particular, we observe that for these different times/profiles, the presence of the $\widetilde{E}_l$ leads to higher $BI_\%$. This behavior can be understood because $v_{shock}$ is lower for these times and the $\widetilde{E}_l$ component is necessary to decelerating ions and reflected them. Without this component, only the magnetic reflection is present and $BI_\%$ has the same amplitude as the previous maximum.

6. Finally, Figure 9 confirms the key role of $E_l$ field in backstreaming ions formation except when approaching $\theta_{Bn} = 45°$ ($N_{Box} = 8 - 9$) while another reflection process is also at work (in absence of $E_l$). This represents an indirect way to stress the noticeable impact of ~~Fermi type process related to~~ the magnetic field in this angular range. This magnetic reflection process is more evidenced at lower $\theta_{Bn}$ since ions need less parallel velocity to be reflected back into the upstream region. This statement can be quantified more precisely as explained in section 5.

This result is an illustration of the impact of the electrostatic component at the shock front. As well known, this component works to decelerate incoming ions (i.e. accelerate in our Solar Wind frame) and to accelerate electrons (i.e. decelerate in our Solar Wind frame) to the downstream region (Savoini and Lembège, 1994; Bale et al., 2005). As a consequence, this electrostatic component reveals to be an essential ingredient in the formation of backstreaming ions, especially ~~at lower $\theta_{Bn}$ where $E_{l\parallel}$ is higher. Obviously, this is not the case for the electrons which are counteracted in their reflection process by the same parallel component.~~ at higher $\theta_{Bn}$.

## 5   Discussions

A previous paper (Savoini and Lembège, 2015) has demonstrated that all reflected ions had suffered the same $\overrightarrow{E} \times \overrightarrow{B}$ drift in the velocity space which can account for the pitch angle distributions observed at the shock front. In fact, the key point is the time spent by particles within the front shock which finally decides whether ions will escape to form the "*FAB*" (with a

pitch angle $\alpha \approx 0°$) or "GPB" populations (with a pitch angle $\alpha \neq 0°$) where $\alpha$ is the angle between the velocity vector and the magnetic field. In other words, the "FAB" population may be associated to a large drift along the shock front (and/or long interaction time) during which particles see a time varying shock front and lose their phase coherency; this case corresponds to a large angular range $\Delta_{int}\theta_{Bn}$ mentioned in section 3.1. In contrast, the ions of the "GPB" population have a shorter interaction time with the shock front associated to a small angular range $\Delta_{int}\theta_{Bn}$. Present test-particle simulations allow to have a deeper insight on the spatial origin of the observed "FAB" and "GPB" populations. Then, we have to analyze more carefully the ion velocity distribution.

Figure 10 plots the local perpendicular velocity distribution functions $f(\overrightarrow{v}_{\perp 1}, \overrightarrow{v}_{\perp 2})$ in both "FCE" and "HE" approaches (where $\overrightarrow{v}_{\perp 1}$ and $\overrightarrow{v}_{\perp 2}$ refer the ion perpendicular velocity components defined with respect to the local magnetic field). All plots are obtained at the end of the simulations and take into account the different populations observed in Figures 4 and 7 (i.e. the both distinct "peaks" of $\theta_{Bn}^{exit}$ angle for lower $N_{box}$ are included); results issued from "FCE" (left panels) and "HE" (right panels) configurations are considered. For the "HE" configuration, we choose an initial time $\widetilde{T} = 4848$ (see Figure 9 for reference) which corresponds to a maximum of $BI_\%$ in order to have enough reflected ions in the velocity space. Results from three different boxes, $N_{Box} = 1, 4$ and $8$, are represented in order to give an overview of the whole ion foreshock components.

Results of the "FCE" configuration (left panels) can be analyzed from Figures 4, 7 and 10. Plots of the $E_l \neq 0$ case (Figure 10) show that the $N_{Box} = 1$ has a low number of upstream reflected ions which leads to a poor statistics and a noisy $f(\overrightarrow{v}_{\perp 1}, \overrightarrow{v}_{\perp 2})$ distribution. Nevertheless, it evidences approximately a distribution with a maximum slightly non centered at $\overrightarrow{v}_\perp = 0$. Then, this distribution can be viewed as a mixing of "GPB" and "FAB" populations, even if the "GPB" population with a pitch angle different from $0$ is the largest one. When moving further into the foreshock region (i.e. lower $\theta_{Bn}$ angle with $N_{Box} = 4$), the number of reflected ions drastically increase and we observe more clearly the characteristic partial ring of the "GPB" population (as in Savoini and Lembège (2015)). In addition, the center of the ring is also partially filled-in because of partial diffusion due to particles having large $\Delta_{int}\theta_{Bn}$ range corresponding to the $\theta_{Bn}^{exit}$ peak around $60°$ in Figure 4 ~~, peak $P_2$~~) and/or by the intrinsic time fluctuations of the front which tends to blur out the velocity distribution both in perpendicular and parallel directions. At last, in agreement with the associated small $\theta_{Bn}^{exit}$ of Figure 4, $N_{Box} = 8$ (in Figure 10) also evidences a non Maxwellian-like distribution ($\alpha \neq 0°$). When we look at the $E_l = 0$ case, we observe roughly the same behavior for all boxes even if the decrease of the backstreaming ions number in box $N_{box} = 8$ makes the comparison difficult.

A further analysis requires a similar approach with the "HE" configuration where we follow a succession of independent expanding shock profiles in order to exclude the impact of the time fluctuations on the velocity distribution $f(\overrightarrow{v}_\parallel, \overrightarrow{v}_\perp)$; then no ion diffusion associated to these fluctuations is allowed. Results of the "HE" configuration (right panels of Figure 10) show reflected ions for $N_{Box} = 1$ when $\widetilde{E}_l \neq 0$ but, once again no reflected ions can be seen when $\widetilde{E}_l = 0$. This evidences the importance of the electrostatic potential wall in order to reflect upstream ions for high $\theta_{Bn}$. On the other hand, if the amplitude of the perpendicular velocity is roughly the same for the two different configurations ("FCE" and "HE"), the "HE" case shows a very well formed ring in contrast with the "FCE" case which exhibits a diffuse velocity space. This illustrates that fields time variations ("FCE" configuration) are much more efficient to diffuse particles than the fields spatial variations ("HE" configuration). Similarly, $N_{Box} = 4$ exhibits a clear ring which is a feature of the "GPB" population, the center of the ring

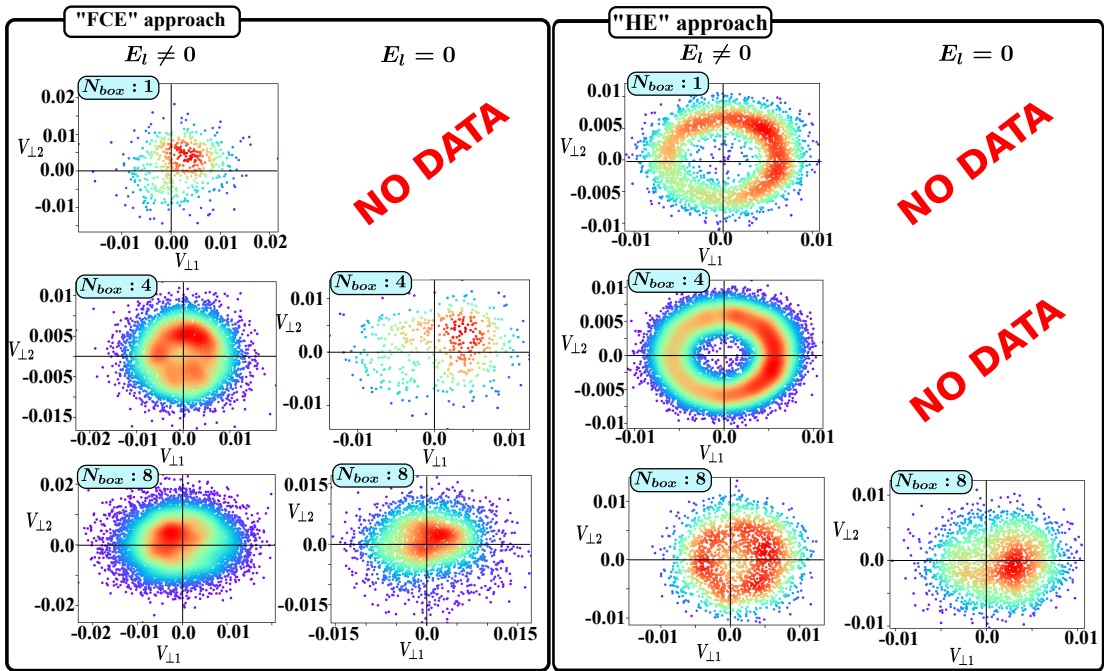

**Figure 10.** Local perpendicular ion velocity space ($v_{\perp 1}$, $v_{\perp 2}$) of all backstreaming ions computed at the end time of the simulations (i.e. after $3\tau_{ci}$ for all simulations) for boxes $N_{Box} = 1, 4$ and 8 in the "*FCE*" approach when the $\widetilde{E}_l$ is included (left panels); the case $\widetilde{E}_l = 0$ is not plotted since the percentage $BI_\%$ is very weak (see Figure 8). The right panels (b) show similar results for the same boxes in "*HE*" configuration (corresponding to $\widetilde{T} = 4848$) in both cases where $\widetilde{E}_l$ is included and artificially excluded; statistical results where $BI_\%$ is too weak are not shown. Red and blue colors hold for maximum and minimum density value in the velocity space.

is not partially filled-in since time velocity diffusion is excluded. These results demonstrate that the formation of the "*FAB*"-
like population is also mainly due to ion velocity diffusion related to the time fluctuations of the shock front which can have different origins as described in previous works (Kucharek et al., 2004; Bale et al., 2005). Finally, for $N_{Box} = 8$, the velocity distribution drastically changes from a ring ($\widetilde{E}_l \neq 0$) to a localized bump ($\widetilde{E}_l = 0$) roughly similar to the "*FCE*" case. ~~In both "*FCE*" and "*HE*" approaches, the absence of the electrostatic component leads to a lower ion reflection efficiency (i.e., ions spend longer time within the shock front) and so, to a more diffuse distribution due to space/time front fluctuations.~~ It is clear
that the number of reflected ions decreases drastically as $\widetilde{E}_l$ field components are artificially suppressed. But, more important, is that the formation of a non gyrotropic distribution does not depend strongly on the $\widetilde{E}_l$ component and is mainly controlled by the convective electric field through the $\overrightarrow{E}_t \times \overrightarrow{B}$ drift in the velocity space as already described in Savoini and Lembège (2015).

Let us remind that each distribution results from a combination of all particles originating from one given box (no matter
where they end up spatially); we can identify it as a "pseudo local" distribution. This differs from the more common strategy

based on measurements of "local" ion distributions as performed in Savoini and Lembège (2015) but which did not precise, at that time, which part of the curved shock front, the FAB and GPB ions are issued from. A further analysis is needed and is left for later work.

## 6    Conclusions

Present test-particle simulations reinforce the scenario described in Savoini and Lembège (2015) and have allowed us to investigate more deeply the formation of the ~~backstreaming ions into the foreshock~~ ion foreshock. ~~With this approach, one can evidence more clearly the impact of the shock front non stationarity on the ion foreshock formation and the role of the $\widetilde{E}_l$ electric component on the two kinds of concerned backstreaming populations (i.e. "GPB" and "FAB" populations).~~ In summary, ~~they have allowed to analyze the dependence of the ion foreshock versus~~ the impact of three quantities/effects has

been identified: (i) the electric ~~ostatic~~ field (separately the $\widetilde{E}_l$ and the $\widetilde{E}_t$ components), (ii) the magnetic field and (iii) the shock front non stationarity. The synoptic of Figure 11 summarizes the importance of each impact versus the shock front curvature from the edge of the ion foreshock ($\theta_{Bn} \approx 70°$) to $\theta_{Bn} = 45°$. In this sketch, the colors vary from strong (full color) to weak (white color) intensity as function of their respective influence on the ion dynamics. These different effects are the following:

1. **Impact of the $\overrightarrow{E}_l$ field on the ion reflection process**. As well-known, the built up potential wall at the shock front

(i.e. the electric field $\overrightarrow{E}_l$ along the shock normal $\overrightarrow{n}$ ) is mainly responsible for the deceleration (i.e. acceleration in our reference frame) of the incoming upstream ions by the shock front. The $\overrightarrow{E}_l$ component has essentially two distinct impacts: (i) without this electric component, no reflected ions are observed for $\theta_{Bn} > 62°$ whereas in presence of this electric component at the edge of the ion foreshock ($\approx \theta_{Bn} \leq 70°$), even one bounce reflection ion can be observed; (ii) at lower angles ($\theta_{Bn} \leq 50°$) many ions are reflected without the help of the $\widetilde{E}_l$ component and can be associated to a

magnetic mirror reflection ~~(i.e. fast Fermi acceleration)~~. Then, in Figure 11, $\widetilde{E}_l$ is only reported as "strong" around the edge of the ion foreshock to emphasize its mandatory action for high $\theta_{Bn}$ angles.

2. **Impact of the $\overrightarrow{E}_t$ field on the ion reflection process**. Figure 10 evidences that the convective electric component $\overrightarrow{E}_t$ is always present in our simulation (we are in the solar wind reference frame and then, $\overrightarrow{E}_t \neq 0$ within the curved propagating shock front). ~~and is responsible for the formation of the "GPB"/"FAB" populations.~~ Our previous work

(Savoini and Lembège (2015)) was only able to show that the $\overrightarrow{E} \times \overrightarrow{B}$ drift scenario in the velocity space foreseen by Gurgiolo et al. (1983) was at the origin of two distinct "GPB" (i.e. one bounce) and "FAB" (i.e. multi-bounces) populations only separated from the particle time history within the shock front.

But this scenario was not able to distinguish the relative importance between the two electric field components $\overrightarrow{E}_l$ and $\overrightarrow{E}_t$ respectively. This question has been clarified in the present paper since *FAB* and *GPB* populations formed by the

$\overrightarrow{E} \times \overrightarrow{B}$ drift in the velocity space are evidenced with ou without the $\overrightarrow{E}_l$ component (Figure 10). Then, the $\overrightarrow{E}_l$ field seems to be less dominant that $\overrightarrow{E}_t$ field. Finally, Figure 11 illustrates the $\overrightarrow{E} \times \overrightarrow{B}$ drift impact by a dark color almost uniform

within the whole quasi-perpendicular region. ~~As a conclusion, acceleration of the two populations is mainly supported by the convective electric component and can be associated to the "*SDA*" or Shock Drift Acceleration.~~

3. **Impact of the $\overrightarrow{B}$ field on the ion reflection process**.

The magnetic field component is important for several reasons: (i) its increase at the shock front "builds up" the $\overrightarrow{E}_l$ component (space charge effect) which reflects back incoming low energy ions and (ii) more importantly, it is also directly responsible for the reflection of some ions (i.e. through the magnetic mirror reflection) and for the drift along the shock front of the multi-bounce ions (i.e. "FAB" population). Then, this population gains energy as ions propagate in the $\overrightarrow{E}_t$ direction along the shock front. Nevertheless, as $\theta_{Bn}$ decreases from $90°$ to $45°$, the ion reflection becomes easier since their parallel guiding center velocity needed to overcome the shock front velocity decreases (Paschmann et al., 1980). This behavior is clearly illustrated herein by the increase of the percentage of reflected ions $BI_\%$ as $\theta_{Bn}$ decreases (i.e., $N_{box} \nearrow$). This behavior persists even in absence of $\overrightarrow{E}_l$ where $BI_\%$ is only reduced by a factor of 2.5 as illustrated by Figure 6). Then, if the magnetic field is important in the whole quasi-perpendicular region, we emphasize in Figure 11 its stronger impact near $\theta_{Bn} \approx 45°$ where it is mainly responsible of reflection (i.e. magnetic mirror) and acceleration (i.e. Fermi type) of ions (Webb et al., 1983).

4. **Impact of the shock front non stationarity**. Present simulations show that the reflection process is not continuous both in time and in space, but strongly depends on the local shock front profile met by incoming ions at their hitting time. This behavior is difficult to be identified in experimental measurements since the particles coming from different shock locations and at different times are mixed; in contrast, this can be easily evidenced in our "*HE*" test particules configuration. This configuration evidences that particular shock profiles are more suitable for the formation of backstreaming ions than other ones. Indeed, we observe modulations of the $BI_\%$ percentages in Figure 9 which are much more pronounced than in our "*FCE*" configuration (Figure 8). These modulations are so strong that $BI_\%$ drops to 0 periodically which means that for certain shock front profiles no ion can escape into the upstream region. This behavior is observed for all $N_{box}$ at the same time (i.e. same shock profile) which implies that the ion reflection does not depend on the location along the shock front but essentially on the global profile of the shock at a given time. In the present simulations, we can identified 4 distinct and noticeable "bursts" (i.e. maxima $BI_\%$ values) with an average cyclic occurrence period of $1\widetilde{\tau}_{ci}^{shock}$ (where $\widetilde{\tau}_{ci}^{shock}$ is defined at the shock ramp). Surprisingly, $N_{box} = 0$ (Figure 9) does not evidence the same "bursts" as the others, which suggests that the time variations of the shock front (and associated particle diffusion as suggested by Kucharek et al. (2004)) are mandatory to obtain backstreaming ions around the edge of the ion foreshock (i.e. high $\theta_{Bn}$). This point will require a further investigation.

In summary, present results show that the formation of the ion foreshock is not a continuous process but must be considered as time dependent, which leads to "bursty" emission of backstreaming ions. Three different contributions have been evidenced: (i) the $\overrightarrow{E}_l$ component in the ion global reflection process in particular for high $\theta_{Bn}$; (ii) the magnetic field $\overrightarrow{B}$ essentially observed when $\overrightarrow{E}_l = 0$ for lower $\theta_{Bn}$ such as the magnetic mirror reflection and (iii) finally, the $\overrightarrow{E}_t \times \overrightarrow{B}$ drift in the velocity

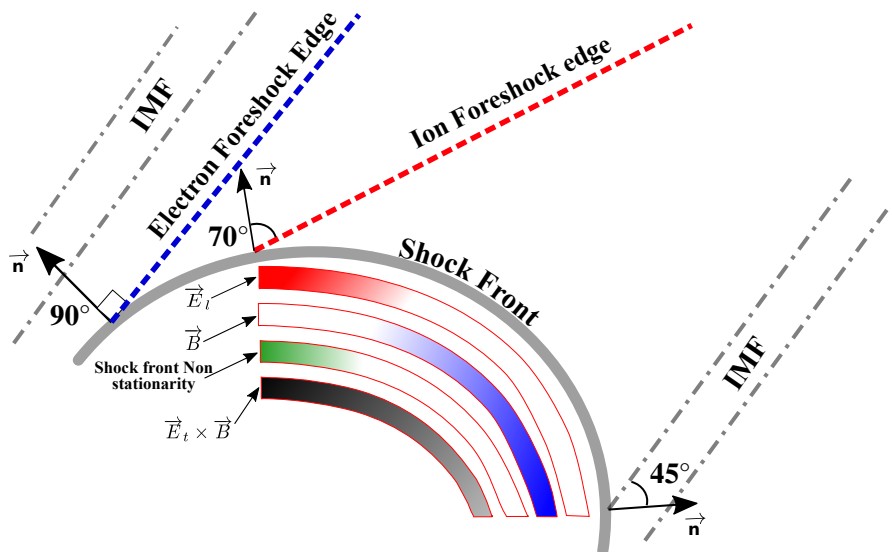

**Figure 11.** Sketch of the ion foreshock in the quasi perpendicular shock region, illustrating the angular areas along the curved front where four main identified processes contributing to and/or impacting the formation of backstreaming ions apply (namely the longitudinal electric field $\vec{E}_l$, the magnetic field, the shock front nonstationary and the convective electric field $\vec{E}_t$). Each process is illustrated by different thick band along the curved front which are shifted one from each other in order to avoid overwhelming the sketch. One color (red, blue, green and black respectively) is associated to each process. The varying intensity of the color indicates where the process is strong (full color) or weak (white color). This allows to identify at a glance the angular areas where the different processes are complementary or accumulating. The upstream interplanetary magnetic field (IMF) is reported in grey (dotted-dashed lines) as well as the shock front itself. Typical directions $\theta_{Bn} = 90°$ and $70°$ (blue and red colored dashed lines) defined between the local shock normal $\vec{n}$ and the IMF indicate the location where the electron and ion foreshock edge initiates from the curved shock front respectively. The electron foreshock edge is indicated as a reference.

space mainly sustained by the convective electric field which is necessary to generate both "*FAB*" and "*GPB*" populations as described in Savoini and Lembège (2015).

     Unfortunately, the impact of the shock front nonstationarity on the ion foreshock is difficult to analyze (see for example Figure 7) for two different reasons: (i) the "*time-of-flight*" effects mix reflected ions coming from different shock profiles and (ii) even if some shock profiles are more efficient than others to reflect ions, their respective impacts disappear rapidly since 500   they are being blurred out by the impact of less efficient profiles on particles as time evolves.

*Author contributions.* P. Savoini and B. Lembege contributed to the design and implementation of the research, to the analysis of the results and to the writing of the manuscript. The data have been produced by P. Savoini

*Competing interests.* There is no competing interests for this paper

*Acknowledgements.* Numerical simulations were performed on the TGCC computer center located at Bruyeres le Chatel (near Paris), which
we thank for its support (DARI project A0050400295). One of the authors (BL) acknowledges the French Centre National d'Etudes Spatiales
(CNES) for its support under APR- W-EEXP/10-01-01-05 et APR-Z-ETP-E-0010/01-01-05 and ISSI (Bern, Swiss) for supporting the col-
laboration network "Resolving the Microphysics of Collisionless Shock Waves ≫. This work also received financial support by the program
"Investissements d'avenir" under the reference ANR-11-IDEX-0004-02 (Plas@Par). Thanks are addressed to Yann Pfau-Kempf and another
referee for helpful comments.

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
