# Peer review of "A Deep Insight into the Ion Foreshock with the help of Test-particle Two-dimensional Simulations"

_Annales Geophysicae, 2020_

## Referee Comment (RC1) · Anonymous Referee #1 · 13 Apr 2020

The authors applied 2D test particle simulations based on field profiles from PIC simulations to understand how a quasi-perpendicular curved shock reflects ions. The authors tested various parameters using a self-consistent field model and a stationary field model that expands in time. The authors determine how electric field, $\theta_{BN}$, and non-stationarity affect the reflection process. Although shocks are one of the fundamental particle accelerators throughout the universe, how shocks reflect ions is still poorly understood. This work provides many detailed results that can significantly improve our understanding. I would like to recommend the paper for publication providing that my concerns below are addressed.

About main conclusions:

I agree with the authors that $E_l$ plays an important role in reflecting ions, but I doubt whether the parallel component of $E_l$ can accelerate ions. $E_l$ causes a potential change across the shock. I agree that $E_l$ causes energy change for ions and electrons that cross the shock as stated in line 330 "this component works to decelerate incoming ions and to accelerate electrons to the downstream region". For reflected ions that do not cross the shock, however, there is no potential change before and after the reflection. I believe that the role of $E_l$ is to build up a potential wall to prevent low energy ions from crossing downstream, right?

The authors claim that $\boldsymbol{E_t} \times \boldsymbol{B}$ drift due to convective electric field is very important. I cannot agree with this statement without mentioning the frame of reference. What about $\boldsymbol{E_t} \times \boldsymbol{B}$ drift in the shock normal incidence frame, the de Hoffmann-Teller frame, or the spacecraft rest frame when observing an earthward IP shock? For example, in the HT frame without $\boldsymbol{E_t}$, the drift in the solar wind rest frame corresponds to the motion of shock surface along the tangential direction. Therefore, as $\boldsymbol{E_t} \times \boldsymbol{B}$ drift is frame dependent, it is important to mention the frame of reference when discussing its role.

Other than $\boldsymbol{E_t} \times \boldsymbol{B}$ drift, there is also grad-B drift. In the shock normal incidence frame, the direction of grad-B drift is along $\boldsymbol{E_t}$ resulting in energy increase, i.e., shock drift acceleration. In the solar wind rest frame, such mechanism can cause velocity increase $dV = 2V_n + 2V_{HT}$ (where $\boldsymbol{V_n}$ is local shock normal velocity in the solar wind rest frame). As B has z component, the grad-B drift direction has XY component. Based on shock drift acceleration model, larger $\theta_{BN}$ results in larger energy increase indicating longer drift distance, which is consistent with Figure 4. Therefore, can grad-B drift at least partially affect $\theta_{BN}^{exit} - \theta_{BN}^{hit}$ as a function of $\theta_{BN}$?

I agree with the authors about the impact of $\theta_{BN}$. However, in the simulation configuration, different $\theta_{BN}$ causes different $M_A$ (from 5 to 3). $M_A$ is also an important factor that can affect BI%. I think the effect of varying $M_A$ needs to be mentioned when discussing the impact of $\theta_{BN}$.

BI% shows burst and drops to 0 periodically in the HE model. I am wondering whether some parameters at the shock front may vary in a similar way, such as the strength of electric field and the gradient of magnetic field.

Other issues:

In Figure 1, there are upstream structures like SLAMS, foreshock cavities, and ULF waves in the FCE model (panel 2b) whereas there is nothing upstream in the HE model (panel 1b). Would upstream structures play a role and cause differences between two models? For example, they may reflect ions back downstream and decrease BI%.

In line 181, I am confused by the term "multi-bounces process". Is this diffusive shock acceleration? Is this "multi-bounces" between the shock surface and upstream structures? Or is this just at the shock surface within one ion gyroradius?

In lines 75-76, the induced electric field is generated by the solar wind. Does this mean that the PIC simulation is not in the solar wind rest frame? In line 126, the induced/convective electric field is due to the relative motion between the solar wind and the shock front. I think the convective electric field should be calculated using the local plasma bulk velocity, right? Or do the authors mean that the convective electric field is transformed from the shock rest frame ($-\boldsymbol{U} \times \boldsymbol{B}$) to the solar wind rest frame using the relative speed between the solar wind and the shock front? $\boldsymbol{E}_t \times \boldsymbol{B}$ is important, but it is unclear how $\boldsymbol{E}_t$ is obtained and difficult for me to check the direction of $\boldsymbol{E}_t$ and $\boldsymbol{E}_t \times \boldsymbol{B}$.

In line 192, $1\tau_{ci} \approx 4\tau_{ci}^{shock}$. Is $\tau_{ci}$ the value in the solar wind? If it is true, the field strength at the middle of the ramp is four times the solar wind field strength. I assume that the field strength at the middle of the ramp is smaller than the downstream field strength meaning that the field strength compression ratio is larger than 4, right?

The time variation of magnetic field of the shock profile can induce electric field. This component of electric field is not included in the HE model. Does this induced electric field play a role?

Figure 11 needs some more text in the conclusion section. For example, does $\vec{B}$ refer to magnetic field or magnetic mirror reflection? How the effect of EXB depends on $\theta_{BN}$ is not discussed in the conclusion section. Does black (white) mean longer (shorter) drift distance or stronger (weaker) effect on the reflection?

Wording problems:

Line 142, when BI% is first mentioned in the main text, I have to go back to the abstract to find its meaning.

I am confused by some terms. It is unclear whether "magnetic mirror reflection (Fast Fermi)", "specular reflection with the conservation of the magnetic moment", "Fermi type reflection", "Fermi type process", "mirror reflection or Fermi reflection", "Fermi type one acceleration process", "fast Fermi acceleration", and "shock drift acceleration" refer to the same process.

In section 2.2, the HE model is first introduced, so I expected to see the results from the HE model first instead of the FCE model in section 3. It may be better to be in the same order.

In line 77, although readers can find magnetic field configuration from the authors' previous papers, it would be better if the authors can simply add an "out-of-plane" symbol and an arrow in Figure 1 to indicate the IMF direction (and perhaps electric field direction at the shock front). Or the authors can at least refer to Figure 11.

In lines 214-215, there are two "in particular" in this sentence. I suggest replacing the second one with "especially".

In line 235, maybe it is better to revise it as "**Figure 7** shows very similar escaping angle distribution compared with Figure 4…"

In line 388, "the impact of the electrostatic field" should be "the impact of the **electric** field" as both components are discussed.

In line 405, impact -> Impact

---

## Referee Comment (RC2) · Yann Pfau-Kempf (Referee) · 4 May 2020

General comments

"A deep insight into the Ion Foreshock with the help of Test-particles Two-dimensional simulations" by Philippe Savoini and Bertrand Lembège presents a detailed analysis of 2D test-particle simulations of the ion foreshock. The simulations are tailored towards clarifying the role of various electric field components and the shock dynamics in the formation of the previously-reported field-aligned beam (FAB) and gyrophase-bunched (GPB) foreshock ion populations. Overall the study is well constructed but the manuscript would greatly benefit from a range of clarifications. I have a series of

comments I would like to see discussed/addressed by the authors, as well as a number of suggestions for technical corrections listed below.

Specific comments

- Lines 22-24: More recent terrestrial foreshock studies could be cited too, but I leave it to the authors to decide as this does certainly not need to be exhaustive. Examples are Strumik et al 2015 (10.1002/2015GL064915); Liu et al 2017 (10.1002/2017JA024480); Otsuka et al 2018 (10.3847/1538-4357/aaa23f); Gutynska et al 2019 (10.1029/2019JA026970); Urbář et al 2019 (10.1029/2019JA026734); Turc et al 2019 (10.1029/2019GL084437). I would also like to draw the attention of the authors to the recent paper by Battarbee et al 2020 (10.5194/angeo-2019-115) which studies ion reflection at the non-stationary terrestrial bow shock, albeit in the quasi-parallel region.

- Lines 73-74 and further: It is unclear from reading this manuscript how the electric field component split is performed and which terms in the equations exactly correspond to El and Et respectively. With respect to what are "transverse" and "longitudinal" defined? I assume this would be the magnetic field but then I am confused in particular by the occurrence of parallel electric field (l. 209) and even El// (l. 236). Even though this has been treated in previous articles I would appreciate if these key elements were introduced here as well as the definition of the various electric field components is a critical piece of information for this study. I am also confused by the notations: is there a difference between components noted with a tilde, an arrow and without?

- The nomenclature regarding Fermi processes is confusing and could be made consistent throughout. Or, if different processes are meant, then they should be introduced in more detail.

- I suggest to add "shock" to "front" on line 57 to avoid potential confusion with the fields in front/upstream of the shock.

- Figures 1, 2 and 11: I would suggest to show the in-plane IMF direction and mention the out-of-plane component of the IMF for clarity.

- Lines 63-64: Section 2 also describes the test-particle simulations. Sections 3-6 exist in the current version so this paragraph should be updated.

- End of section 2.1: The system size, spatial resolution and scalings are pieces of information that would be useful, in particular since they are being referred to, e.g. line 118 or Figure 9.

- Section 2.2: I would also suggest to order the HE and FCE consistently throughout (abstract, introduction, figure 1, section 2.2, sections 3 and 4), maybe indeed taking first FCE and then HE every time.

- Lines 109-110: Is $v\_thi$ averaged over the box?

- Figure 2 and lines 111-112: The particles are colour-coded differently in the rest of the manuscript so this mention of the colouring of particles is incorrect.

- End of section 2.2. The description of the HE procedure is unclear to me. - Where is the origin of the transformation? Are only shock points transformed or everything outside of the origin? Are the resulting fields then interpolated back to the original resolution or is the grid resolution expanding as well? - Lines 131-132: "Then, each front profile is selected within a same simulation time range DT..." What does this mean? Is DT $\sim$ 4 the difference between 5.4 and 1.2 cyclotron times and there is 174 "snapshots" taken to propagate 1 million particles each? Section 4 mentions 100 runs, also, so I guess that only the more interesting last 100 are taken? - Reading Section 4 lines 284-289 I understand better. So that paragraph and the one in section 2.2 should maybe be joined with an effort to clarify the scheme. - How long are the HE and FCE runs? - When are the test particles released in the FCE case? At a single time or over a certain period?

- Figure 3 and 6b: Is the colour code a density? What are the units? "Spatial distribution

of percentage" is rather imprecise. It would also help the comparison if all panels were on the same colour scale, maybe with a logarithmic colour scale overall as Box 1 is very different.

- Figure 4 and discussion in the text from line 164 onwards: What is the definition of the shock position in this study? As shown by Battarbee et al 2020 (10.5194/angeo-2019-115), Figure 2 in particular, depending on the criterion taken the "position" of the shock can vary dramatically.

- Figure 5: It might help the comparison if all plots had the same y axis, maybe with a logarithmic scale?

- Figure 8: It might help the comparison if all plots had the same y axis. Line 258 and the figure: are the blue lines linear fits or drawn "by hand" to illustrate?

- Line 261: How are the particles released at the same distance if they were released within the boxes of Figure 2?

- Line 279: I object to the use of fully self-consistent here as this is about test particles.

- Figure 9, lines 285 and 289: The text mentions 3 gyroperiods, the figure caption says 10. Which is correct?

- Figure 10: - In the caption, the case not plotted is for Box 1, as others are shown in the left panel. There is on (b) in the figure so that can be removed from the caption too. Can the colour scale be clarified? Is it a derived phase-space density? - It would be good to clarify also in the text: are these distributions a combination of all particles originating in one box, no matter where they ended up spatially? Could the authors illustrate/discuss the impact of this, as opposed to taking the distribution in a given spatial region, which is the more common strategy?

Technical corrections

- The title capitalisation is inconsistent.

- Line 1: "test-particle" (no -s)

- Line 7: on/off; detailed

- Line 18: This copyright statement is incompatible with the license granted at top of each page and on the discussion web page.

- Line 45: A large scale

- Line 48: "until 2 RE" or "up to 2 RE"; RE/Earth radius has not been introduced yet.

- Line 49: First occurrence of E and B, they could be introduced here.

- Line 51: loses

- Line 72: technique

- Line 82 and elsewhere: Alfvén

- Figure 1: time independent; in the fully consistent expansion model

- Line 113 & 116: boxes

- Line 123: a homothetic transformation

- Line 160: depending on

- Line 201: Do you mean "it cannot be necessary" or "it could be unnecessary"? I guess the latter.

- Line 246: stationary

- Figure 8: The lines are black and not blue. And both cases are switched, so black/red are respectively with and without El.

- Line 275: in a previous paper

- Line 277: correspondence

- Line 282: " missing

- Line 305: followed by

- Line 306: corresponds to a half gyration

- Line 316: taking into account

- Line 320 & 322: the peak amplitude

- Line 322: Fermi

- Figure 9: developed

- Line 330: As is well known

- Line 331: Lembege; "As a consequence" or "Consequently"

- Line 340: during which particles see (no comma); corresponds

- Line 341: mentioned

- Lines 343 and 402: discriminate

- Line 347: accelerates

- Figure 10: "(see Figure 8)" (no "to")

- Line 363 and 367: f1 resp. f2 and not P1, P2, I believe.

- Line 368: At last (?)

- Line 370: look at; roughly

- Line 384: No "Then"

- Line 388: dependence

- Line 397: associated to a

- Line 403: components

- Line 408: extra )

- Line 429: such as

- Line 432: respective

- Line 433: since they are being blurred

- Figure 11: "black" instead of "dark", maybe?

- Line 437: produced

---

## Author Comment (AC1) · 22 Jun 2020

A deep insight into the Ion Foreshock with the help of Test-particles Two-dimensional simulations" by Philippe Savoini and Bertrand Lembege

**Answer to the comments of referee #1;**

**We thank the referee for the helpful comments. Please find below our detailed answers to each comment which are indicated in bold letter. Corrections have been directly inserted in the text in blue color and sentences and/or parts of the sentences to be suppressed are also indicated.**

The authors applied 2D test particle simulations based on field profiles from PIC simulations to understand how a quasi-perpendicular curved shock reflects ions. The authors tested various parameters using a self-consistent field model and a stationary field model that expands in time. The authors determine how electric field, $\theta BN$, and non-stationarity affect the reflection process. Although shocks are one of the fundamental particle accelerators throughout the universe, how shocks reflect ions is still poorly understood. This work provides many detailed results that can significantly improve our understanding. I would like to recommend the paper for publication providing that my concerns below are addressed.

About main conclusions:
I agree with the authors that $El$ plays an important role in reflecting ions, but I doubt whether the parallel component of $El$ can accelerate ions. $El$ causes a potential change across the shock. I agree that $El$ causes energy change for ions and electrons that cross the shock as stated in line 330 "this component works to decelerate incoming ions and to accelerate electrons to the downstream region". For reflected ions that do not cross the shock, however, there is no potential change before and after the reflection. I believe that the role of $El$ is to build up a potential wall to prevent low energy ions from crossing downstream, right?

**Yes, as indicated in the text, authors agree with the referee concerning the role of the electric potential wall which decelerates ions and reflects back low energy ions and some modifications have been made in the paper to be more precise.**
**Such reflection process is done with energy conservation when the potential amplitude is the same before and after the reflection (i.e. the total work of the electric force is null). Nevertheless, this scenario is only valid if the ions during their reflection "see" exactly the same shock profile (i.e. the shock profile has to be constant in time with a planar geometry) which is not the case here where both curvature effects and shock front non stationarity are included. In fact, the ion reflection process in this paper can be classified as follows :**
**1) Case 1: ions suffer a one bounce reflection in a short time. These ions are classified as "GPB" and can be associated to a Fermi type reflection. In this case, the role of the potential wall is *limited* to the reflection and not to the acceleration of the ions because ions see roughly the same shock profile both in time and in space.**
**2) Case 2: ions suffer a drift along the shock front and may suffer multi-bounces before being reflected back into the upstream region. These ions can stay a long/very long time (several local ion gyro-periods) within the shock front. In this case, the theory of a constant electric potential (both in time and space) is not valid anymore and its difference**

between the time/space when ions hit and leave the shock front can be related to the ion acceleration parallel to the magnetic field.

**The authors have clarified the role of the electric field in the text (see the beginning of the section 3.2)**

The authors claim that $Et \times B$ drift due to convective electric field is very important. I cannot agree with this statement without mentioning the frame of reference. What about $Et \times B$ drift in the shock normal incidence frame, the de Hoffmann-Teller frame, or the spacecraft rest frame when observing an earthward IP shock? For example, in the HT frame without $Et$, the drift in the solar wind rest frame corresponds to the motion of shock surface along the tangential direction. Therefore, as $Et \times B$ drift is frame dependent, it is important to mention the frame of reference when discussing its role.

**The authors have modified the text accordingly. Nevertheless, it is important to point out that in presence of a curved propagating shock, it is not possible to define a global de Hoffman-Teller frame in our case. We remind at different locations of the text that we are in the solar wind reference frame, and the field Et herein is carried by the expanding shock front itself.**

Other than $Et \times B$ drift, there is also grad-B drift. In the shock normal incidence frame, the direction of grad-B drift is along $Et$ resulting in energy increase, i.e., shock drift acceleration. In the solar wind rest frame, such mechanism can cause velocity increase $dV = 2Vn + 2VHT$ (where $Vn$ is local shock normal velocity in the solar wind rest frame). As B has z component, the grad-B drift direction has XY component. Based on shock drift acceleration model, larger $\theta BN$ results in larger energy increase indicating longer drift distance, which is consistent with Figure 4. Therefore, can grad-B drift at least partially affect $\theta exit - \theta hit$ as a function of $\theta_{BN}$?. I agree with the authors about the impact of $\theta BN$.

**Thanks to the referee. Yes, the paper was not clear enough concerning this problem, and the text has been modified in order to clarify our approach. The main goal of the paper is to investigate the possible source of the ion energy gain when ions are reflected back by the shock front. In a previous paper, we have evidenced that both "FAB" and "GPB" could have the same origin, namely a ExB drift in the velocity space present at the shock front. The goal of the present paper is to go deeper and to analyze not the source of these two populations but how ions are accelerated within the shock front before backstreaming into the upstream region. Then, it is important to split this mechanism into two distinct parts:**

**1) the first one coming from the ExB drift which "forms" the two backstreaming populations "GPB" and "FAB" (our previous paper). Of course it is important to retrieve both with present test particles simulations.**

**2) the second related to the acceleration of particles themselves. We evidence two distinct processes : (i) the El_para which can accelerate ions along the magnetic field and the Et field component which can accelerate (multi-bounce) ions along the shock front (these ions suffer a gradBxB drift along the shock in the same direction of the Et field and then are accelerated by this electric field). We have clarified this point in the text.**

**In addition, when the Et (convective electric field coming from the plasma moving frame – aka shock front) is artificially suppressed NO ions are reflected anymore. Then, it is not possible for us to analyze the reflection process in this case. This behavior evidences that the convective electric field is mandatory to observe the ion reflection and then, is definitely more important than the grad//B force (i.e. mirror magnetic reflection) in this case.**

However, in the simulation configuration, different $\theta BN$ causes different MA (from 5 to 3). MA is also an important factor that can affect BI%. I think the effect of varying MA needs to be mentioned when discussing the impact of $\theta BN$.

**In our geometric configuration (curved shock wave), the expanding shock decelerates in time and then, also the Mach Number from 5 to 3. Nevertheless, as evidenced in figure 1 panels 2a and 2b, the shock front can be described by an approximate circle which evidences the low dependency of MA (i.e. shock velocity) versus the $\theta BN$ angle.**

BI% shows burst and drops to 0 periodically in the HE model. I am wondering whether some parameters at the shock front may vary in a similar way, such as the strength of electric field and the gradient of magnetic field.

**Even when the BI% drops to 0, the general shape of the shock front is unchanged. No such strong variations are observed correspondingly in the magnetic and electric fields amplitudes.**

Other issues:
In Figure 1, there are upstream structures like SLAMS, foreshock cavities, and ULF waves in the FCE model (panel 2b) whereas there is nothing upstream in the HE model (panel 1b). Would upstream structures play a role and cause differences between two models? For example, they may reflect ions back downstream and decrease BI%.

**The modulations observed in the quasi-perpendicular upstream region (fig. 1, panel 2b) are mainly due to (i) a propagating whistler wave and (ii) a small turbulence associated to the electron foreshock. Both fluctuations have small amplitudes and shorter time/space variations in comparison to the characteristic ion scales. Then, we did not observe any impact on the backstreaming ion dynamics (No backstreaming ions are reflected back towards the shock front).**
**In addition, we are interested by the reflection process itself (by defining and analyzing $\Theta_{hit}$ and $\Theta_{exit}$ quantities) which occurs exclusively within the shock front and is independent of upstream fluctuations. Then, we concluded that these upstream fluctuations have no impact on the reflection process studied in our paper.**

In line 181, I am confused by the term "multi-bounces process". Is this diffusive shock acceleration? Is this "multi-bounces" between the shock surface and upstream structures? Or is this just at the shock surface within one ion gyroradius?

**This paper is the extension of a previous study [Savoini et Lembege, 2015] where ion trajectories have been extensively studied. Diagnosis evidence that multi-bounces ions stay within the shock front and are not between upstream structures and the front. Nevertheless,**

we have to point out that the convective electric field present in the upstream region in the common shock reference frame is in fact present within the shock front region in the present Solar Wind reference frame. For this reason, we can argue that the ExB drift is the most important mechanism in order to account for our observations concerning the origin of the "GPB" and "FAB" populations (especially the convective Et field component in this drift).

The goal of this paper is focused on the backstreaming ions origin and not on the study of the acceleration process in term of SSA or SDA processes. We think that this specific study will need deeper investigation which is left for a further work. All associated sentences have been removed from the paper. Moreover, we have removed the term "process" which was confusing.

In lines 75-76, the induced electric field is generated by the solar wind. Does this mean that the PIC simulation is not in the solar wind rest frame?

**No, the referee is right, the simulations are in the Solar Wind frame. The text was unclear and has been modified.**

In line 126, the induced/convective electric field is due to the relative motion between the solar wind and the shock front. I think the convective electric field should be calculated using the local plasma bulk velocity, right? Or do the authors mean that the convective electric field is transformed from the shock rest frame ($-U \times B$) to the solar wind rest frame using the relative speed between the solar wind and the shock front? $Et \times B$ is important, but it is unclear how $Et$ is obtained and difficult for me to check the direction of $Et$ and $Et \times B$.

**The induced electric field is NOT computed in these simulations from the relation –UxB but obtained self-consistently in the previous self-consistent PIC simulation directly from the Maxwell's equations. Let us remind that, since we use a spectral PIC code, we can identify separately transverse field Et and longitudinal (space effects) electric field El. In both configurations "FCE" and "HE", we can analyze these two distinct components El and Et independently. Obviously, the induced electric field corresponds to the convective electric field Et which is associated to the propagation of the curved shock front into the SW plasma. The direction of the Et field is along the curved shock front. We have modified Figure 1 in order to indicate the configuration used in this paper. The ExB drift is only responsible to the formation of the "GPB" and "FAB" populations as described in our previous paper (Savoini et Lembege, 2015). Unfortunately, by construction, it is not possible to define a global reference frame where we can cancel this field (a frame propagating with the same velocity that shock front). This difficulty is mainly due to the propagation of the curved shock wave into the Solar Wind in all directions. For this reason, we are not able to cancel artificially the Et component (as the El component) since that would mean that we had to "stop" the shock front which should be totally unrealistic.**

**The authors have modified the text in order to precise the definition of the Et component introduced in the present test-particle simulations (see section 2.1).**

In line 192, $1\tau \approx 4\ \tau_{ci}^{shock}$. Is $\tau_{ci}$ the value in the solar wind? If it is true, the field strength at the middle of the ramp is four times the solar wind field strength. I assume that the field strength at

the middle of the ramp is smaller than the downstream field strength meaning that the field strength compression ratio is larger than 4, right?

**Yes, the overshoot is about 7 but the exact value depends not only on both the angle theta_Bn and the time because of the front nonstationarity, but also on the Mach number which decreases as the shock propagates. Nevertheless, the value of B = 4 measured in the middle of the ramp corresponds to the local value averaged over the time range under consideration in the present simulations (the upstream value is Bo=1.5).**

The time variation of magnetic field of the shock profile can induce electric field. This component of electric field is not included in the HE model. Does this induced electric field play a role?

**As mentioned above, our PIC simulations are done with a spectral code (i.e. Maxwell and Poisson equations are resolved in the Fourier space) which allows us to separate the electrostatic component (El in Poisson's equation) and the electromagnetic component induced by the time variation of the magnetic field, named Et (from Ampere equation). Then, even in the HE configuration, we can (in fact, we have to) include the Et component in order to follow the propagating shock front and we observe its impact in both FCE and HE configurations.**

Figure 11 needs some more text in the conclusion section. For example, does *B* refer to magnetic field or magnetic mirror reflection? How the effect of EXB depends on $\theta BN$ is not discussed in the conclusion section. Does black (white) mean longer (shorter) drift distance or stronger (weaker) effect on the reflection?

**The authors have clarified the description of this figure in the text.**

Wording problems:
Line 142, when BI% is first mentioned in the main text, I have to go back to the abstract to find its meaning.

**A more precise definition has been now introduced.**

I am confused by some terms. It is unclear whether "magnetic mirror reflection (Fast Fermi)", "specular reflection with the conservation of the magnetic moment", "Fermi type reflection", "Fermi type process", "mirror reflection or Fermi reflection", "Fermi type one acceleration process", "fast Fermi acceleration", and "shock drift acceleration" refer to the same process.

**All these sentences refer to the same process (i.e. magnetic reflection) but we use also the term Fermi type reflection or even Fast Fermi since an energy gain of the reflected particle (i.e. a Fermi type acceleration) is associated to this reflection process while the shock front propagates. We have simplified and replaced most of these terms by magnetic reflection in the text, only in conclusion we introduce the term Fermi acceleration.**

In section 2.2, the HE model is first introduced, so I expected to see the results from the HE model first instead of the FCE model in section 3. It may be better to be in the same order.

**Thanks to the referee. The authors have changed the text accordingly, so that FCE case is introduced and described first, and HE case follows after**

In line 77, although readers can find magnetic field configuration from the authors' previous papers, it would be better if the authors can simply add an "out-of-plane" symbol and an arrow in Figure 1 to indicate the IMF direction (and perhaps electric field direction at the shock front). Or the authors can at least refer to Figure 11.

**A new plot 1a (Figure 1) has been added showing a perspective view of the simulation plane in order to clarify the shock geometry.**

In lines 214-215, there are two "in particular" in this sentence. I suggest replacing the second one with "especially"
**Done**

In line 235, maybe it is better to revise it as "**Figure 7** shows very similar escaping angle distribution compared with Figure 4..."
**Done**

In line 388, "the impact of the electrostatic field" should be "the impact of the **electric** field" as both components are discussed.
**Done**

In line 405, impact -> Impact
**Done**

---

## Author Comment (AC2) · 22 Jun 2020

General comments
"A deep insight into the Ion Foreshock with the help of Test-particles Two-dimensional simulations" by Philippe Savoini and Bertrand Lembege presents a detailed analysis of 2D test-particle simulations of the ion foreshock. The simulations are tailored to- wards clarifying the role of various electric field components and the shock dynamics in the formation of the previously-reported field-aligned beam (FAB) and gyrophase- bunched (GPB) foreshock ion populations. Overall the study is well constructed but the manuscript would greatly benefit from a range of clarifications. I have a series of comments I would like to see discussed/addressed by the authors, as well as a number of suggestions for technical corrections listed below.
Specific comments

- Lines 22-24: More recent terrestrial foreshock studies could be cited too, but I leave it to the authors to decide as this does certainly not need to be exhaustive. Examples are Strumik et al 2015 (10.1002/2015GL064915); Liu et al 2017 (10.1002/2017JA024480); Otsuka et al 2018 (10.3847/1538-4357/aaa23f); Gutynska et al 2019 (10.1029/2019JA026970); Urbar et al 2019 (10.1029/2019JA026734); Turc et al 2019 (10.1029/2019GL084437). I would also like to draw the attention of the authors to the recent paper by Battarbee et al 2020 (10.5194/angeo-2019-115) which studies ion reflection at the non-stationary terrestrial bow shock, albeit in the quasi- parallel region.

**The authors thanks the referee for the references. We have inserted some of them (the others are not inserted since not directly related to the main topics developed in the present paper).**

- Lines 73-74 and further: It is unclear from reading this manuscript how the electric field component split is performed and which terms in the equations exactly correspond to El and Et respectively. With respect to what are "transverse" and "longitudinal" defined? I assume this would be the magnetic field but then I am confused in particular by the occurrence of parallel electric field (l. 209) and even El// (l. 236). Even though this has been treated in previous articles I would appreciate if these key elements were introduced here as well as the definition of the various electric field components is a critical piece of information for this study. I am also confused by the notations: is there a difference between components noted with a tilde, an arrow and without?

**The authors have clarified these terms in section 2.**
**Notations: tilde refer to normalized quantities used herein (issued from the PIC simulations); arrows refer to general vector quantities; no arrow refer to general scalar quantities.**

- The nomenclature regarding Fermi processes is confusing and could be made con- sistent throughout. Or, if different processes are meant, then they should be introduced in more detail.

**The authors agree with the referee and have replaced "Fermi process", by "magnetic reflection" in the whole paper because all these sentences refer to the same process (i.e. magnetic reflection). The terms "Fermi type reflection" or even "Fast Fermi" have been used since an energy gain of the reflected particle (i.e. a Fermi type acceleration) is associated to this reflection process while the shock front propagates. We have now simplified and replaced most of these terms by "magnetic reflection" in the text, only in conclusion we introduce the term Fermi acceleration.**

- I suggest to add "shock" to "front" on line 57 to avoid potential confusion with the fields in front/upstream of the shock.
**Done**

- Figures 1, 2 and 11: I would suggest to show the in-plane IMF direction and mention the out-of-plane component of the IMF for clarity.
**Done, to see new panel in Figure 1 .**

- Lines 63-64: Section 2 also describes the test-particle simulations. Sections 3-6 exist in the current version so this paragraph should be updated.
**In order to clarify, « general » features of PIC simulations and of test particles are shortly reminded in Sections 2.1 and 2.2 respectively. All results of « HE » model are now in Section 4, including (i) specific features of « HE » model (which is new) which are detailed in Sec 4.1, and (ii) corresponding results which are collected in Sec. 4.2.**

- End of section 2.1: The system size, spatial resolution and scalings are pieces of information that would be useful, in particular since they are being referred to, e.g. line 118 or Figure 9.
**Done in section 2**

- Section 2.2: I would also suggest to order the HE and FCE consistently throughout (abstract, introduction, figure 1, section 2.2, sections 3 and 4), maybe indeed taking first FCE and then HE every time.
**We have modified the text accordingly.**

- Lines 109-110: Is v_thi averaged over the box?
**V_thi is just the standard deviation of a Maxwellian and we populate each box with 100 000 particles obtained randomly following this function. The number of particles is high enough to describe correctly this function.**

- Figure 2 and lines 111-112: The particles are colour-coded differently in the rest of the manuscript so this mention of the colouring of particles is incorrect.
**We have clarified this point in the figure caption.**

- End of section 2.2. The description of the HE procedure is unclear to me. - Where is the origin of the transformation? Are only shock points transformed or everything outside of the origin? Are the resulting fields then interpolated back to the original res- olution or is the grid resolution expanding as well?
          **We have clarified this point.**

- Lines 131-132: "Then, each front profile is selected within a same simulation time range DT..." What does this mean? Is DT ~ 4 the difference between 5.4 and 1.2 cyclotron times and there is 174 "snap- shots" taken to propagate 1 million particles each? Section 4 mentions 100 runs, also, so I guess that only the more interesting last 100 are taken?

**We have clarified this point.**

- Reading Section 4 lines 284-289 I understand better. So that paragraph and the one in section 2.2 should maybe be joined with an effort to clarify the scheme.

**We have clarified this point.**

- How long are the HE and FCE runs? - When are the test particles released in the FCE case? At a single time or over a certain period?

**We use always the time T=1,2 \tau_ci as the beginning of our simulation and launch the test-particles at this single time. Then, we run the simulation until the time 5,4 \tau_ci; then we repeat this same procedure for each shock  profile.**

- Figure 3 and 6b: Is the colour code a density? What are the units? "Spatial distribution of percentage" is rather imprecise. It would also help the comparison if all panels were on the same colour scale, maybe with a logarithmic colour scale overall as Box 1 is very different.

**The number of reflected ions is so low that the density per grid-cell can be too low to a nice representation. So, we used a gaussian interpolation which gives the relative density weight of each ion. Then, the color code (vertical bar) gives only an indication of the relative density amplitude. We have clarified the figure caption accordingly.**

- Figure 4 and discussion in the text from line 164 onwards: What is the definition of the shock position in this study? As shown by Battarbee et al 2020 (10.5194/angeo- 2019-115), Figure 2 in particular, depending on the criterion taken the "position" of the shock can vary dramatically.

**The paper's Battarbee simulates a quasi-parallel shock wave which has a much more complicated structure (and very different) than a quasi-perpendicular shock. In our paper, we follow a quasi-perpendicular shock (more "simple" profile) where it is easier and more precise to identify the middle of the shock front, i.e. from the ramp which is used for defining the shock front location.**

- Figure 5: It might help the comparison if all plots had the same y axis, maybe with a logarithmic scale?

**The authors use linear scales in Figure 4 and 5 in order to emphasize the differences between the two distinct distribution functions (i.e. one bounce and multi-bounces ion populations).**

- Figure 8: It might help the comparison if all plots had the same y axis. Line 258 and the figure: are the blue lines linear fits or drawn "by hand" to illustrate?

**The blue lines are only for illustration and effectively have been drawn "by hand". We have clarified this point in the figure caption of Figure 8.**

- Line 261: How are the particles released at the same distance if they were released within the boxes of Figure 2?

**Thanks to the referee. The sentence has been corrected.**

- Line 279: I object to the use of fully self-consistent here as this is about test particles.

**The simulations are "self-consistent" in terms of field evolution because fields are issued from a full self-consistent PIC simulation but the authors agree that the term fully is not the appropriate term and has been removed.**

- Figure 9, lines 285 and 289: The text mentions 3 gyroperiods, the figure caption says 10. Which is correct?

**10  was a mistake and has been corrected; 3 gyroperiods is correct**

- Figure 10: - In the caption, the case not plotted is for Box 1, as others are shown in the left panel. There is on (b) in the figure so that can be removed from the caption too. Can the colour scale be clarified? Is it a derived phase-space density?

**The "(b)" has been removed and we have added the Box_1 in the figure. This plot has been obtained as Figure 3 by using a Gaussian interpolation. Then, as for Figure 3, only an indication of the relative density amplitude in the velocity space is shown.**

- It would be good to clarify also in the text: are these distributions a combination of all particles originating in one box, no matter where they ended up spatially? Could the authors illustrate/discuss the impact of this, as opposed to taking the distribution in a given spatial region, which is the more common strategy?

**Let us remind that each distribution results from a combination of all particles originating from one given box (no matter where they end up spatially); we can identify it as a "pseudo local" distribution. This differs from the more common strategy based on measurements of "local" ion distributions as performed in Savoini et Lembège (2015) but which did not precise, at that time,  which part of the curved shock front,  the FAB and GPB ions are issued  from. A deeper investigation is necessary to clarify this point and is out of scope of this paper. This point is clarified at the end of the section 5.**

Technical corrections
- The title capitalisation is inconsistent.
**Done**
 - Line 1: "test-particle" (no -s) - Line 7: on/off; detailed
**done**

- Line 18: This copyright statement is incompatible with the license granted at top of each page and on the discussion web page.
**We have removed it**
- Line 45: A large scale
**done**
- Line 48: "until 2 RE" or "up to 2 RE"; RE/Earth radius has not been introduced yet.
**done**
- Line 49: First occurrence of E and B, they could be introduced here.
**We have improved the sentence**
- Line 51: loses
**done**

- Line 72: technique

**done**

- Line 82 and elsewhere: Alfvén

**done**

- Figure 1: time independent; in the fully consistent expansion model

**done**

- Line 113 & 116: boxes

**done**

- Line 123: a homothetic transformation

**done**

- Line 160: depending on

**done**

- Line 201: Do you mean "it cannot be necessary" or "it could be unnecessary"? I guess the latter.

**done**

- Line 246: stationary

**done**

- Figure 8: The lines are black and not blue. And both cases are switched, so black/red are respectively with and without El.

**done**

- Line 275: in a previous paper

**done**

- Line 277: correspondence

**done**

 - Line 282: " missing

**done**

- Line 305: followed by

**done**

- Line 306: corresponds to a half gyration

**done**

- Line 316: taking into account

**done**

- Line 320 & 322: the peak amplitude

**done**

- Line 322: Fermi

**done**

- Figure 9: developed

**done**

- Line 330: As is well known

**done**

- Line 331: Lembege; "As a consequence" or "Consequently"

**done**

- Line 340: during which particles see (no comma); corresponds

**done**

- Line 341: mentioned

**done**

- Lines 343 and 402: discriminate

**done**

- Line 347: accelerates

**done**

- Figure 10: "(see Figure 8)" (no "to")

**done**

- Line 363 and 367: f1 resp. f2 and not P1, P2, I believe.

**P1 and P2 have been removed from the whole paper**

- Line 368: At last (?)

**done**

- Line 370: look at; roughly

**done**

- Line 384: No "Then"

**done**

- Line 388: dependence

**done**

- Line 397: associated to a

**done**

- Line 403: components

**text have been changed**

- Line 408: extra )

**text has been changed**

- Line 429: such as

**done**

- Line 432: respective

**done**

- Line 433: since they are being blurred

**done**

- Figure 11: "black" instead of "dark", maybe?

- Line 437: produced

**done**